

# Introduction of a Trans-scale Numerical Simulation Framework Focusing on Urban Boundary Layer: WOCSS V1.0

Wei Li[1,2], Shuo Leng[3], Sunwei Li[4*,], Zhenzhong Hu[4,5], Pakwai Chan[6]

[1]Key Laboratory of Far-shore Wind Power Technology of Zhejiang Province, China
[2]PowerChina Huadong Engineering Corporation Limited, China
[3]Department of Civil Engineering, Tsinghua University, China
[4]Institute for Ocean Engineering, Shenzhen International Graduate School, Tsinghua University, China
[5]Institute for Ocean Engineering, Tsinghua University, China
[6]Hong Kong Observatory, 134A Nathan Road, Kowloon, Hong Kong

*Correspondence to*: Sunwei Li (li.sunwei@sz.tsinghua.edu.cn)

**Abstract.** In the field of geoscience, the meso-scale tool to conduct weather forecast, which is also termed as Numerical Weather Prediction (NWP) package, is commonly used for simulating the urban boundary layer in the scale of 1km~100km. In the field of wind engineering, the Computational Fluid Dynamic (CFD) simulation tool is most popular for investigating
the urban wind environment at the scale of 1m~1km. In the present study, a novel framework, named WOCSS with the version v1.0, combing the meso-scale NWP package of the Weather Research and Forecast (WRF) model and the micro-scale OpenFOAM code is introduced thanks to an open-source package of PreCICE. In detail, PreCICE realizes the trans-scale simulation of the urban wind environment through one-way nesting of porting the meso-scale simulation results to the boundaries of the micro-scale simulation. To this end, the adaptions made to the open-sourced codes of WRF and
OpenFOAM are articulated, which fulfil the information exchanges between WRF and OpenFOAM via PreCICE library. A case study concerning the urban wind environment in a residential quarter in Shenzhen, China is conducted using WOCSS V1.0. The case study demonstrates that the proposed framework successfully presents the detailed wind environment inside the residential quarter under realistic meteorological condition.

## 1 Introduction

The simulation of the atmosphere is the key to understand the meteorological, or geoscience, processes that either facilitate the development or poses challenges for the safety of human society. Especially for the simulation of atmospheric flows inside the urban boundary layer, the accurate and reliable numerical model is of importance in the fields of urban planning, disaster mitigation, emergency response and air pollution assessments. Since the outbreak of the COVID-19 in 2020, the air ventilation problem has attracted attentions from scholars in various fields, and the use of a numerical model to predict air
ventilation around the site of interest becomes common practices. Besides, the exploitation of wind energies inside the urban canopy by small-sized turbines installed in high-rise buildings also requires a good tool for predicting the urban wind



environment. More importantly, the numerically simulated wind flow, governed by the complex interactions between atmospheric boundary layer and the urban morphology, could lead to groundbreaking scientific findings in the geoscience field regarding the meteorological phenomenons due to human activities.

As for the numerical simulation of the urban boundary layer, the mainstream tool in the geoscience field is the meso-scale model, which is employed to deliver the numerical prediction of the wind flow at the city scale. Among meso-scale models, which are also widely known as the Numerical Weather Prediction (NWP) package, the Weather Research and Forecast (WRF) model is one of the most popular choices (Kwok and Ng, 2021). In fact, the WRF model provides an effective and efficient computing platform that integrates advances in physics, numeric, and data assimilation to predict the wind flow at a

scale usually in the order of 10~100 km (Skamarock, et al., 2019). More specifically, the WRF model is used to numerically discern the urban wind flow at the city scale (Zhang, et al., 2022) to the block scale (Zhen, et al., 2019). It is noted that the WRF simulation of the urban boundary layer produces not only the wind flow, but also temperatures, humidity and precipitation etc., which is helpful in the assessment of the impacts of various meteorological processes to the human society. Previous studies have been carried out to compare simulation results of the WRF model with field observations and

confirmed the reliability of this widely-used meso-scale NWP software (Muñoz-Esparza, et al., 2018; Mughal, et al., 2020). Other than the meso-scale NWP, the micro-scale model, i.e. the Computational Fluid Dynamics (CFD) simulation code, is also utilized in the investigation of the urban wind environment at the scale of buildings (Blocken, 2014; Weerasuriya, et al., 2018). Governed by the Navier – Stokes equations, CFD simulations yield the details of the wind flow, limited by the resolution of the computational grid, inside and around buildings. As the academic research concerning the fluid dynamics

often relies on various numerical tools, open-source has become a trend in the development of CFD software. Among them, OpenFOAM is the most well-known brand (Jasak, 2009). Instead of comprehensive solvers, OpenFOAM actually provides a toolset for the user to assemble their own solver in order to deal with their specifically defined problems, including turbulence mechanism discovery (Robertson, et al., 2015) and complex diffusion problem (Cheng and Lin, 2022). Hence, the features of OpenFOAM is easy to modify, which supports the development of custom functionality and integration with

other codes. Different from the meso-scale code of the WRF model, the CFD simulation yields mainly the prediction of wind flows. Additional information concerning other meteorological variables, such as temperature, humidity and concentration of certain pollutants, can be obtained from adding modules to the common CFD solvers.

Integrating the advantages of meso-scale and micro-scale simulation tools to understand the wind flow within the complex urban morphology is clearly beneficial in terms of assessing pedestrian comfort, coping with air pollution and securing

public health threatened by airborne disease. For example, Leng et al. (2022) have set up a joint numerical simulation chain using the outputs from the meso-scale model to drive the steady-state CFD run, which improves the prediction of the air quality condition around building clusters. In addition, the integration of the meso-scale and micro-scale is also used in the assessment of wind resources and aerodynamics of wind farms (Gopalan, et al., 2014). Due to the variety of the codes corresponding to both the meso-scale and micro-scale models, there is, to the best of the authors' knowledge, no common

framework available to run the meso-scale and micro-scale simulation in parallel, in which the data are exchanged implicitly



along with both the simulation runs. In order words, there is no commonly available "glue code" to integrate the existing meso-scale model with the micro-scale model running in parallel.

In terms of integrating the meso-scale and micro-scale numerical tool, there currently are several different approaches under development. While it is suggested the CFD code can be up-scaled to simulate the flow at a geophysical scale (Oguro, et al., 2008; Blocken., et al., 2015), there are also practices that utilizes the NWP code to run simulations with grids down to the sub-building scale (Yamada and Koike, 2011). In addition, various micro-scale meteorological models are developed to simulate the wind flow and temperature fields at the sub-building scale (Huttner and Bruse 2009). It is understandable that up-scaling the CFD code to simulate the geophysical flow requires tremendous computational resources and long simulation time, which makes it impractical other than pure academic research. For example, the CFD simulation targeting the urban boundary layer of the Great Tokyo Bay area used more than 9 billion cells, and hence can only be run at supercomputer clusters (Ashie and Kono, 2011). The down-scaled simulation of a NWP model, or the simulation conducted by a micro-scale meteorological model, often parameterize, to some extent, the obstacle influences of the urban morphology. More specifically, the well-established models of ENVI-met (Huttner and Bruse 2009) and MITRAS (Schlünzen, et al., 2003, Salim, et al., 2018) use the equal-spaced grid, and therefore the obstacle effects of buildings are simulated using the mask approach. In other words, the mesh accurately depicts the geometry of the buildings, especially the building envelope in the vertical direction, as shown by Fig. 3 in the work of Kadaverugu, et al. (2021) can not be constructed at an acceptable computational cost. Additionally, the boundary and initial conditions of wind velocities employed by the micro-scale meteorological model often ignores their spatial variations (Giersch, et al., 2022). In other words, the independent use of the micro-scale meteorological model presents an uniform and constant background wind environment. Since both the up-scaling of CFD codes and down-scaling of the NWP to the micro-scale associate with respective shortcomings in the simulation of the urban boundary layer, the joint simulation runs the CFD and NWP codes in parallel could be the solution at present.

Generally, there are one-way and two-way nesting for coupling the meso-scale model and micro-scale model for running the joint simulation. In the category of the one-way nesting, the simulation results from a meso-scale tool are used as the initial and boundary conditions for the micro-scale model. Employing the micro-scale meteorological model of PALM (Maronga, et al., 2015, 2020), Lin, et al. (2021) and Kadasch, et al. (2021) developed the interfaces to connect with the meso-scale model of WRF and COSMO respectively. It is noted that such interfaces are only used to translate or to superimpose synthesized turbulence on the simulated fields from the meso-scale model, and hence lacks the capability to managing the meso- and micro-scale models running in parallel. Different from such approaches, Bakhoday-Paskyabi, et al. (2022) showed the micro-scale wind field with the joint simulation coupling WRF and PALM models targeting an offshore wind farm in Germany. The joint simulation interpolates the WRF results to the boundaries of the PALM domain at the output frequency of the WRF model (10 minutes). Considering the gird size of the micro-scale domain (~10m), such a configuration implies a quasi-steady state simulation at the micro-scale side, which is equivalent to the "snapshot" method employed by Leng et al. (2019). Using the similar method, McRae, et al. (2020) and Kadaverugu, et al. (2021) coupled the



meso-scale model of WRF with the micro-scale meteorological model of ENVI-met and OpenFOAM, respectively, to present the sub-building scale wind fields found in San Jose, California and Nagpur City, India. Their simulations hence showed the wind field at the micro-scale in a quasi-steady state, which lack the temporal variations corresponding to the spatial resolution of ~1m at the micro-scale side. Other than the "snapshot" approach, Schlünzen, et al.(2011) and Zajaczkowski, et al. (2011) suggested to integrate the temporal varying simulation results from the meso-scale model with

the CFD code. For example, Vogel, et al (2022) nested the meso-scale model of WRF with the micro-scale model of PALM-4U. This one-way nesting simulation spatially interpolates the output from the meso-scale model not only at the boundaries but also in the interior of the micro-scale PALM-4U domain. In the category of two-way nesting, the simulations are conducted in parallel by both the meso-scale model and the micro-scale model, and the results are exchanged at the prescribed moments to adjust both sides of the joint simulation. While the CFD simulation is bounded by the results from the

NWP at the boundaries, the CFD results adjust the NWP simulation at the meso-scale grid points via the "objective analysis" process. Although it is acknowledged that the two-way nesting generally outperforms the one-way nesting for the long duration simulation case (Sprague. and Satkauskas, 2015), the computational cost of the two-way nesting simulation is considerably higher (Baklanov and Nuterman, 2002). Moreover, the complex data exchange could collapse the joint simulation run in a two-way nesting mode. In the investigation focusing primarily on the urban wind environment, the meso-

scale simulation results at the city scale is often not the emphasis, and therefore the benefits brought by the two-way nesting scheme are limited. In conclusion, the study on the urban wind environment could be based on the joint simulation of the meso-scale and micro-scale model coupled through the one-way nesting mechanism.

    The present study proposes a framework integrating the meso-scale tool of WRF and the micro-scale tool of OpenFOAM via the open-sourced multi-scale multi-physics coupling tool of PreCICE, named WRF-OpenFOAM Coupled Simulation

System (WOCSS). While the wind flow at the scale of city blocks is simulated by the WRF model, the results output by the WRF model is transferred to initialize and bound the simulation at sub-building scale conducted by OpenFOAM. In fact, the model is built upon the work reported by Leng, et al. (2022). The so-called "snapshot" method is utilized in the simulation of Leng, et al. (2022), in which the wind field simulated by the WRF model within a user-specified time window is averaged to drive a steady-state simulation of OpenFOAM. The present study, on the other hand, introduce a framework to manage the

parallel running of the WRF and OpenFOAM codes, and to exchange meso-scale simulation results to the micro-scale model in a transient manner. In other words, the quasi-steady assumption, which is based on time scale difference between the meso-scale and micro-scale models, explicitly or implicitly employed in previous simulations (Leng, et al. 2022; McRae, et al., 2020; Kadaverugu, et al., 2021) is no longer used in WOCSS. The joint simulation is carried out in two steps. In the first step, the WRF model is run alone to provide the boundary and initial conditions for the fine-scale WRF model in the second

step. The WRF model is then activated in the second step again with the Large Eddy Simulation (LES) scheme enabled for the domain with a fine grid of the spatial scale of ~10m. Also in the second step, the CFD simulation run by OpenFOAM is employed in parallel with the WRF run, which takes the temporally varying output from the WRF simulation as the boundary condition. Since the framework proposed in the present study essentially implements the one-way nesting scheme,





the data exchanged at the coupling interface is stipulated by the WRF model alone and handled by the PreCICE. In other words, the boundary conditions of the CFD code are specified by the WRF simulation results and updated according to the time step of the meso-scale WRF model. Therefore, the CFD simulation supplements the WRF model in WOCSS to provide the detailed flow structures within the micro-scale domain in the simulation interval of the WRF model.

After the introduction, section 2 details the framework for the WOCSS, which includes the descriptions of WRF model, OpenFOAM model, PreCICE interface and the architecture of WOCSS. Section 3 articulate the measures suggested by the present study to process the simulation data from the meso-scale model for integrating into the micro-scale simulation. A case study is present in section 4 to illustrate the set-ups and results of the WOCSS simulation. Conclusions are given in section 5.

## 2 The Framework for the WOCSS simulation

The WOCSS utilizes the open-sourced PreCICE library to deal with the data exchange and joint simulation management, and hence introduces a new framework to run the joint simulation integrating WRF and OpenFOAM. More specifically, the WOCSS contains modifications to the source codes of WRF and OpenFOAM to communicate with the PreCICE, and hence propose a new mechanism to run the WRF and OpenFOAM simulation in parallel while keeping the information from both participants exchanged in the real time. WOCSS is publicly available from the well-established git repositories of Gitee and GitHub (SunweiTsinghua, 2023). In fact, Fig. 1 shows the flow diagram of running the WOCSS simulation.



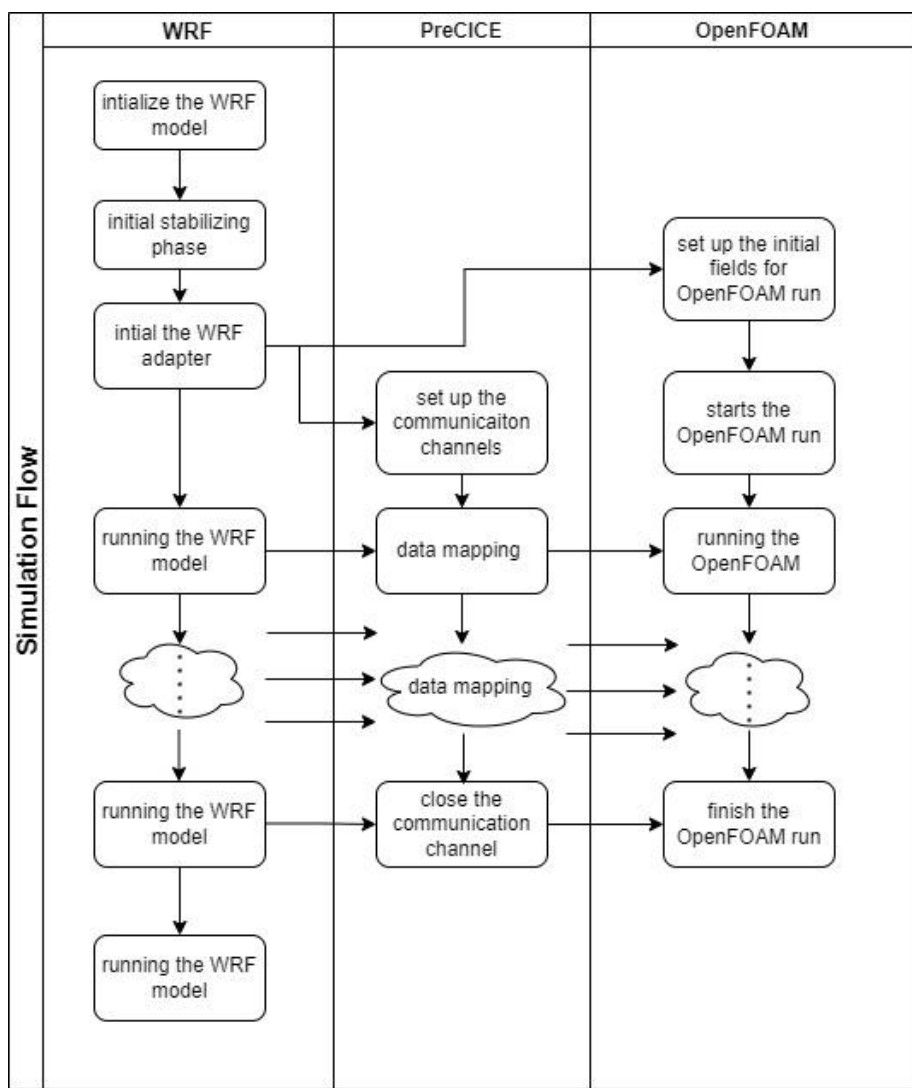


Figure 1: the simulation flow diagram of the proposed framework.

## 2.1 WRF model

The WRF model is a well-known NWP tool to do the weather forecasting, and is hence widely employed in the investigation of the wind patterns in the geoscience field. The WRF model was developed by a collective partnership of the National
Center for Atmospheric Research, the National Oceanic and Atmospheric Administration (represented by the National Centers for Environmental Prediction and the Earth System Research Laboratory), the U.S. Air Force, the Naval Research Laboratory, the University of Oklahoma, and the Federal Aviation Administration since the 1990s. Through integrating recent scientific discoveries in the field of meteorology and geoscience (Marzban, et al., 2020) , the WRF model has been under continuous development through efforts from the community. The version 4 of the WRF model has now been moved





to the well-organized open-source community of GitHub[1], which is easier for the non-experts making contributions to the development of new features.

Based on the templates provided by the original WRF model, in the realization of the so-called "time series" functionality, the authors implement an "adapter", using the term defined by the PreCICE (Chourdakis, et al., 2021), corresponding to the WRF model to take the responsibility to communicate with the PreCICE, and further to the CFD code. The "time series"

functionality of the original WRF model is used to output selective variables at a few designated locations far more frequent than the regular results-saving. The "time series" output from the WRF model are mainly used to compare the simulation results with the observations at weather stations in order to evaluate the reliability of certain WRF simulation runs. The authors expand such output functionality to cover the coupling interface at which the WRF "adapter" is communicating with the PreCICE. In detail, specific meteorological variables, selected by the user, at every simulation time step of the WRF

model are sent to the PreCICE via the developed adapter at the coupling interface.

In the WOCSS, the WRF "adapter" developed by the authors currently calculates and sends the wind velocities, air pressures, temperatures, turbulent kinetic energies and their dissipation rates to the PreCICE. While some variables, such as the wind velocities, are simulation variables of the WRF model itself, and hence are sent to the PreCICE directly, the variables in association with turbulence, such as the dissipation rate of turbulent kinetic energies, are calculated according to the basic

simulation variables of the WRF model. In fact, when the non-local Planetary Boundary Layer scheme of Mellor-Yamada-Janjic (MYJ) scheme (Suelj and Sood, 2010) is invoked in the WRF simulation, the turbulent kinetic energies within the atmospheric boundary layer are directly available, and their dissipation rate can be estimated according to the turbulent kinetic energies and the turbulence length scale, which is also a simulation variable of the MYJ Planetary Boundary Layer scheme. In conclusion, the adapter to the WRF model takes care of calculating and outputting the specific variables to the

PreCICE, and the PreCICE takes care of the rest, including the data mapping and information exchanges, and make it ready for the use of the CFD code.

## 2.2 OpenFOAM model

OpenFOAM is the commonly accepted software for running the CFD simulation at the scales crossing from engines to buildings (Regalado-Rodriguez and Militello, 2022; Bay, et al., 2022). The OpenFOAM code was developed originally by

Henry Weller when he worked in the Imperial College London since 1989. The intention of developing OpenFOAM is to provide the community with an open-sourced tool to operate and solve the partial differential equations. Starting from 2004, OpenFOAM has been released under the General Public License. Currently, there are two branches in developing OpenFOAM, namely the org version maintained by the OpenFOAM foundation and the com version maintained by the ESI Corporation. Both branches are now open-sourced, and widely employed in fields of flow dynamics research and

engineering applications. Instead of providing the standard solvers for specifically targeted problems, the OpenFOAM

---

[1] https://github.com/wrf-model/WRF



actually provides a library of codes capable of solving the well-defined partial differential equation via the Finite Volume Method. Based on such a library, OpenFOAM also provides a series of "standard" solvers targeting the CFD simulation for general purposes.

The PreCICE library provides an "official" adapter to the OpenFOAM code, which takes care of communicating with the
PreCICE in a fluid-structure interaction/heat transfer/fluid-fluid interaction scenarios. The authors simply modified the official adapter to the OpenFOAM model to take in more variables, such as the turbulence variables, when integrating WRF and OpenFOAM models. More specifically, the Fluid-Fluid interaction coupling module of the official adapter is modified in WOCSS to exchange the temperature, turbulent kinetic energy and its dissipation rates with the PreCICE. Given the variables added to the official adapter are contained in any CFD solver with mainstream two-equation turbulence model, the
modification simply translates the temperature, turbulent kinetic energy and dissipation rate to the names understood by the target OpenFOAM solver.

In the proposed framework integrating the WRF and OpenFOAM, the computational domain of the OpenFOAM model is always specified as a box, whose corner coordinates and height are specified by the user. In detail, in the configuration of the adapter to the WRF model, there are specifications of the corner longitudes and latitudes defining the OpenFOAM
computational domain and hence the PreCICE coupling interfaces. In the specification, users have the option to include the lid of the box. Although the lid, which takes the specific meteorological variables at a certain height from the WRF simulation to bound the CFD run, presents a comprehensive coupling effect, it sometimes affects the numerical stability of the CFD run. Regardless of whether the lid is included in the setting-ups of the WRF model with PreCICE coupling functionality enabled, the height of the coupling box should be specified to instruct the adapter the level up to which
outputting the simulation variables.

## 2.3 PreCICE coupling library

The key of the trans-scale simulation done by WOCSS is the data exchange and running management for both meso-scale and micro-scale model. In the WOCSS, the recently emerge package, called PreCICE (Chourdakis, et al., 2021), is employed to take care of the coupling details when running the WRF model and OpenFOAM model in parallel. PreCICE is the newly
developed, open-sourced library for running the partitioned simulation of multi-scale, multi-physics problems. In detail, the PreCICE is capable of linking the existing solvers/codes, which take care of a subset of the target problem, to run the comprehensive and realistic simulation in a coupled way. The library is developed in the groups of Benjamin Uekermann (Usability and Sustainability of Simulation Software) and Miriam Schulte (Simulation of Large Systems) at the University of Stuttgart and in the group of Hans-Joachim Bungartz (Scientific Computing in Computer Science) at the Technical
University of Munich, and provides a convenient way to deal with information exchanges among participants and data mapping at the coupling interface.

The behaviour of the PreCICE is highly configurable in terms of both information exchange schemes and data mapping at the coupling interface. In fact, the PreCICE employs a configuration file, which is written in the XML format, to control



various aspects of the simulation coupling scheme between the participants. In the current version of the WOCSS, the one-way nesting is chosen for coupling the WRF and OpenFOAM codes, which implies the series coupling scheme with explicit data exchange (using the terms of PreCICE configures[2]). Other than the coupling scheme, configurations supported by the PreCICE package, in theory, works with the WOCSS, which includes all settings for data mapping schemes and information exchange monitoring. It is noted that the two-way nesting approach is currently under development based on the four dimensional data assimilation functionality of the original WRF model (Gopalakrishnan and Chandrasekar, 2022). Such an improvement on the adapter to the WRF model will be made publicly available in the version upgrade of WOCSS.

**2.4 The trans-scale simulation flow**

For the trans-scale simulation, the WRF model is initialized first, and the wind, pressure and turbulence fields simulated by the WRF model are sent to the PreCICE interface after the initial stabilizing phase. Using the WRF outputs at the PreCICE coupling interface as boundary conditions of wind velocities, pressures and turbulent variables, the OpenFOAM simulation runs independently from the WRF model when the PreCICE library serves as the communication channel.

The first step to run the trans-scale joint simulation is to specify the coupling interface for both the CFD code and the WRF model, through a designated control file of the adapter (see the appendix for details). The definition of the coupling interface for the OpenFOAM simulation is based on the boundaries taking in the variables from the WRF simulation. In configuring the WRF adapter, the PreCICE coupling interface is defined as a box, whose dimensions are prescribed in longitudes and latitudes pairs of corners and the height. Moreover, users have the control over the variables exchanged through the PreCICE interface. Currently, only wind velocities, temperatures, air pressures, turbulent kinetic energies and dissipation rates are supported by the adapter. In the configuration of the OpenFOAM run, the interfaces are defined as, naturally, the boundaries of the computational domain. The condition of such boundaries is specified as "interface", which allow the PreCICE to update their values according to the simulation results from the WRF run.

Running the WRF and OpenFOAM model independently, which implicitly initializes the PreCICE library, is the second step for the trans-scale simulation. In addition, the WOCSS uses the WRF simulation results after the initial stabilizing phase to set up the initial condition for the OpenFOAM run. More specifically, the adapter to the WRF model is capable of outputting the entire fields of specific variables within the box defined by the PreCICE interface precisely at the moment specified by the user. Afterwards, OpenFOAM utilizes the field data output from the WRF adapter to initialize its own simulation run, which in most cases could reduce the time required for the OpenFOAM run to yield results compatible with the WRF model. In the trans-scale simulation integrating the WRF and OpenFOAM, the data exchange and mapping to make the two models communicable are taken care by the PreCICE in a silent way. The configurations of the PreCICE library, except for the coupling scheme, are generally supported in the current version of WOCSS, which includes the settings corresponding to data mapping and the monitoring for information exchange at the coupling interface.

---

[2] https://precice.org/configuration-coupling.html



Post-processing of the trans-scale simulation is composed of three components, which can be recognized as the third step. Understandably, the post-processing targets the simulation results from the WRF model, the OpenFOAM code and the PreCICE. While the WRF simulation is unaltered in the trans-scale simulation, the results corresponding to the OpenFOAM model and the PreCICE coupling show the coupling effects. Especially for the OpenFOAM simulation, its results show the combined influences of the detailed urban morphology and meso-scale meteorology simulation results, which is the main
target for the trans-scale simulation employing the one-way nesting scheme. In addition to the OpenFOAM simulation results of wind velocities, air pressures and turbulence characteristics, the PreCICE also outputs the data at the coupling interface, mainly for the purpose of debugging. The PreCICE outputs, however, constitute a major component of the trans-scale simulation, which helps to assess the isolated influence of the meso-scale model to the ultimate results of the trans-scale simulation.

## 3 Data Processing and Exchange in the Trans-scale Simulation

In the trans-scale simulation, the simulation variables exchanged at the coupling interface are specified in the control file of the PreCICE, and the current version of WOCSS takes the wind velocity, air pressure, temperature, turbulent kinetic energy and its dissipation rate to exchange between the WRF and OpenFOAM. In the one-way nesting joint simulation, the simulation variables from the WRF model at the coupling interface needs to be processed before porting to the OpenFOAM
code. While the wind velocity, air pressure and air temperature require interpolation, turbulent kinetic energy and its dissipation rate require estimation and correction before interpolation.

### 3.1  Data interpolation

Because the corners of the PreCICE interface in the WRF model may not always align with its gird points, and the WRF simulated variables only available at the grid points, interpolations are necessary to calculate the exchange data at the
coupling interface. In the present study, the horizontal bi-linear interpolation is implemented to calculate the desired exchange data at the interface. More specifically, the four WRF simulated variables at the grid points surrounding the PreCICE interface are interpolated horizontally to have the target variable. For such interpolations, the information on the horizontal locations of the surrounding grids are used to calculate the interpolation weights in the bi-linear interpolation, and the corresponding WRF simulated variables at the current time step are multiplied by the weights, and then to exchange to
the PreCICE.

The initialization of the OpenFOAM simulation, on the other hand, handles interpolation at the OpenFOAM side of the joint simulation. In detail, the WRF simulation results cover the OpenFOAM computational domain are directly ported to the OpenFOAM simulation by the adapter to the WRF model. The simulation variables of wind velocities, air pressures and turbulence variables ported from the WRF model are expected to significantly reduces the time required for the OpenFOAM
simulation to achieve a compatible state with the WRF simulation. Since only the raw results of the WRF model are ported,



the authors coded a new utility in the framework of the OpenFOAM to take care of the three dimensional interpolation using the relatively coarse data from the WRF simulation to specify detailed wind and pressure fields for the relative fine OpenFOAM mesh. Specifically, the adapter to the WRF model not only output the simulation variables, but also the connectivity information of the WRF grid to instruct the new utility of the OpenFOAM to do the three dimensional interpolation. Fig. 2 shows the effect of the utility to convert the WRF simulated wind and turbulent kinetic energy fields into the initial condition of the wind velocity and turbulent kinetic energy (TKE) in the OpenFOAM simulation.

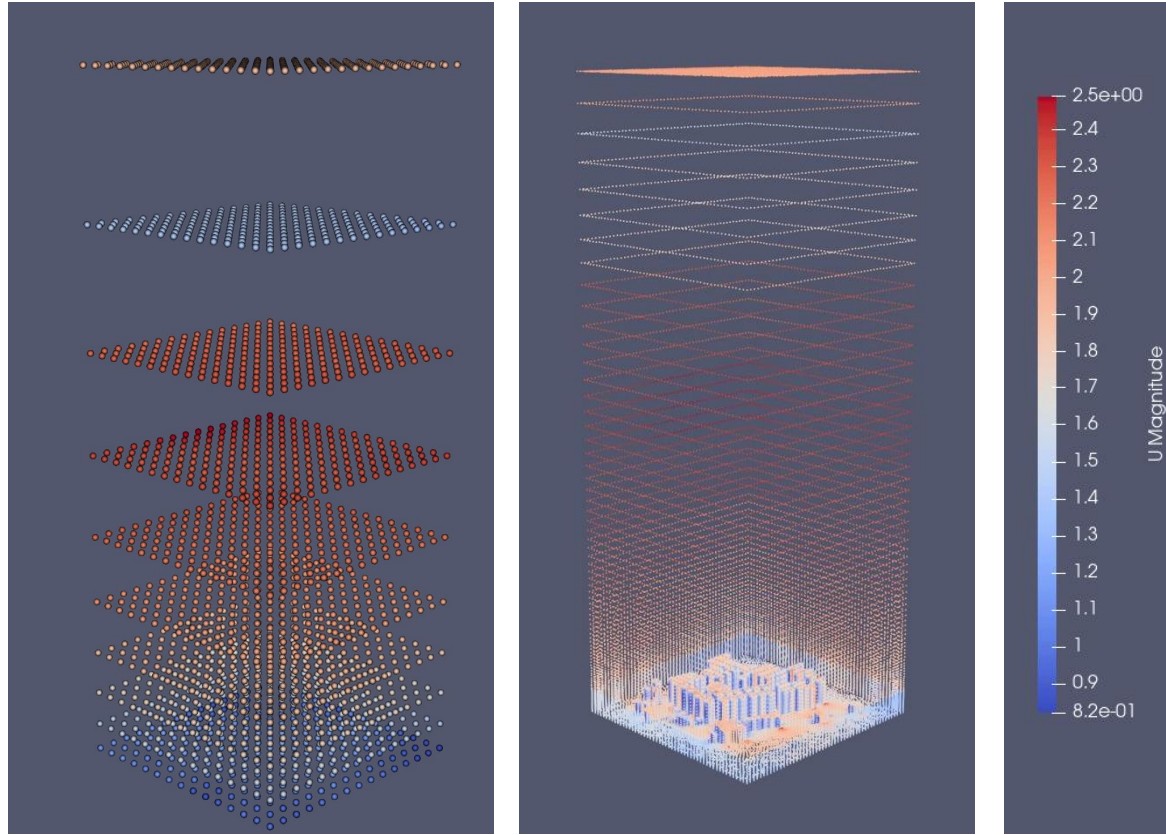

(a)Wind velocity from WRF          (b)Wind Velocity initialized in OpenFOAM



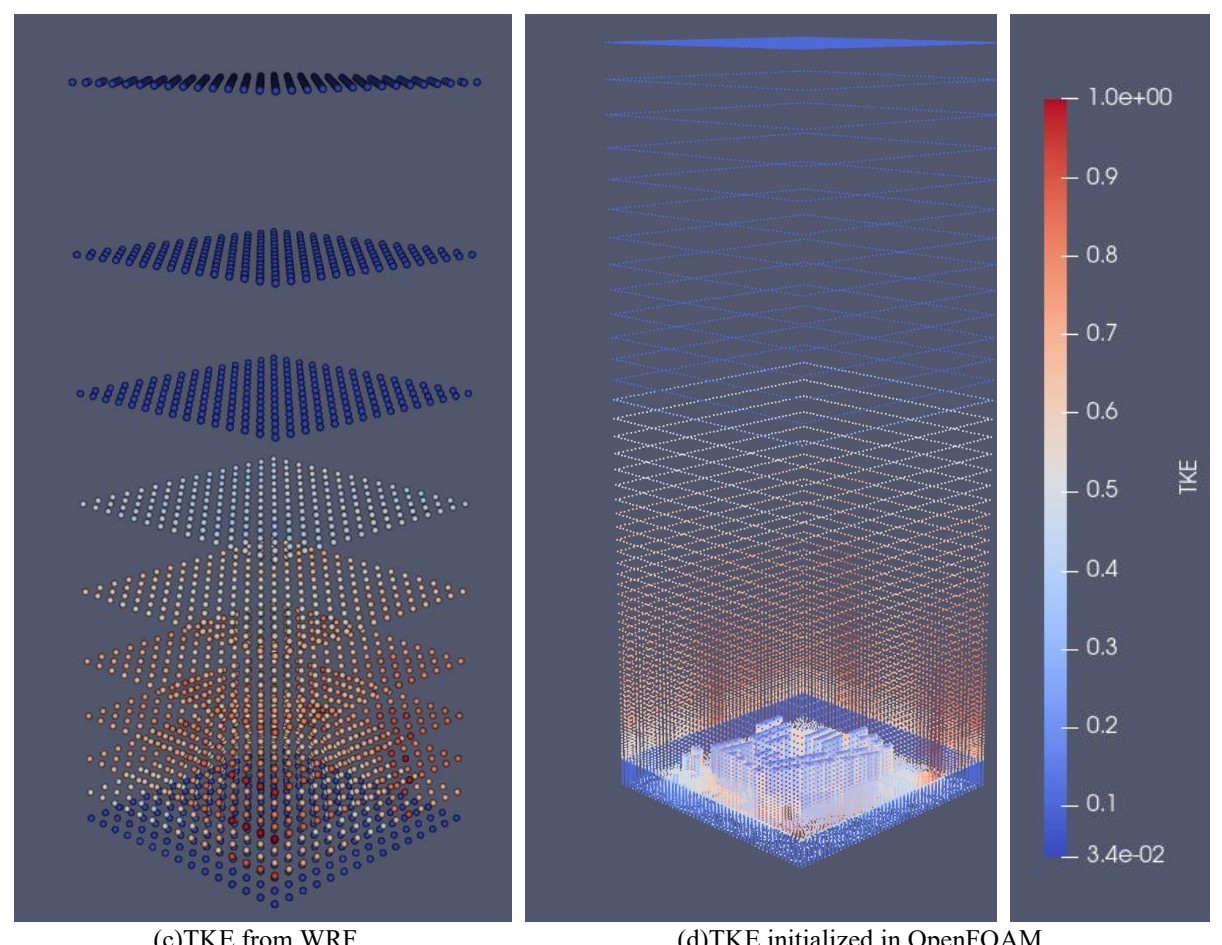

(c)TKE from WRF         (d)TKE initialized in OpenFOAM

**Figure 2 the wind velocity magnitude and turbulent kinetic energy simulated by WRF and the the field interpolated to initialized the OpenFOAM simulation**

**295  3.2 Turbulence Estimation**

Due to the difference for the codes of WRF and OpenFOAM in dealing with the turbulence, it is the responsibility of the WOCSS to handle the "translation" of the turbulence variables given by the WRF model. More specifically, while the turbulent kinetic energy contained in the WRF model is used directly to instruct the OpenFOAM simulation of turbulence, its dissipation rate, which is lacked in the WRF code, is estimated using the turbulent kinetic energy and its corresponding

length scale. In detail, the turbulent kinetic energy and length scale, output from the WRF model with specific Planet Boundary Layer scheme below the height of the OpenFOAM computational domain, are used to calculate the dissipation rate as,



$$\varepsilon = C_\mu^{3/4} \left(\frac{k}{l}\right) \tag{1}$$

In equation (1), $\varepsilon$ is the dissipation rate of the turbulent kinetic energy $k$, $l$ is the turbulence length scale and $C_\mu$ is the empirical constant specified in conventional $k - \varepsilon$ two equation turbulence model, which takes the value of 0.09 in the present study and can be adjusted by users. It is noted that the simulation of turbulent length scales output from the WRF model are close to zero at the lowest level of the grid, which leads to erroneous estimates of the dissipation rate close to the ground. In detail, the small value of the turbulent length scale leads to unrealistically large results of dissipation rate according to equation (1), and that would be problematic when initializing the simulation at the micro-scale scale. Consequently, the present study proposes to estimate the turbulent length scale below 10m as,

$$l = \kappa z \tag{2}$$

In equation (2), $\kappa$ is the Karmann constant, which takes the value of 0.4, and $z$ is the height from the ground. Therefore, the dissipation rate of turbulent kinetic energy is determined as piece-wise function of the height and the primary part of the atmospheric boundary lay shows the dissipation rate according to the WRF simulated turbulent kinetic energy and corresponding length scale as calculated by equation (1).

## 4 Case Study

The tans-scale joint simulation of the WOCSS is carried out for a case study showing urban wind environment in Shenzhen, China. More specifically, the simulation presents the wind environment around a residential quarter with low-rise to median-rise buildings. The geographic locations of the WOCSS simulation target area is shown in Fig. 3.

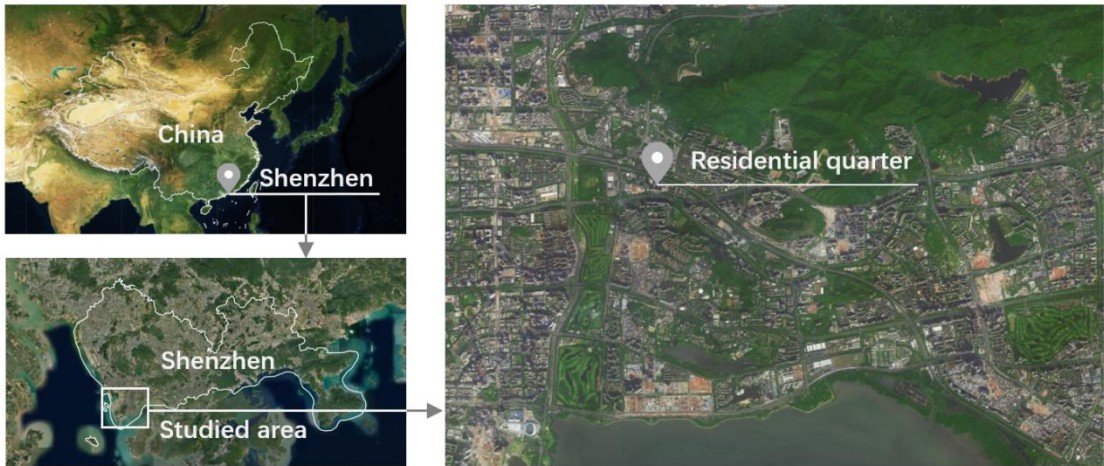

**Figure 3 The geographic locations of the case study area, displayed on top of © Google Maps 2022**



In the case study, the boundary and initial conditions of the trans-scale simulation are extracted from the reanalysis data (the well-known ERA5 datase[3]) made available by the European Centre for Medium-range Weather Forecasts. The ERA5 dataset provides hourly estimates of atmospheric, land and oceanic meteorological variables at a horizontal resolution of 30km and covers the atmosphere up to the height of 80km. Through integrating the vast amount of historical observational data, the ERA5 dataset improves the estimates from a global numerical model to provide reliable estimates of meteorological

variables in history. In addition to the initial and boundary conditions for the meso-scale simulation run, the micro-scale simulation requires detailed geometric information of the buildings within the target area. These geometric data are collected from Baidu Map[4] , which is a comprehensive geographic information system application. Baidu Maps provides worldwide building information services, and has the most detailed data in China. A crawler developed by the authors is used to extract information of the target buildings from Baidu Maps web pages. The acquired data is originally organized as vector

geographical information system files, including the bottom outline and height of the building. The geometries of the target buildings are then ported into the well-accepted plain text format of json for the use of the Gmsh[5], a wide-used mesh generator for the CFD simulation. The geometric shapes of the buildings in the target area on top of the satellite image provided by Baidu Map is shown in Fig. 4

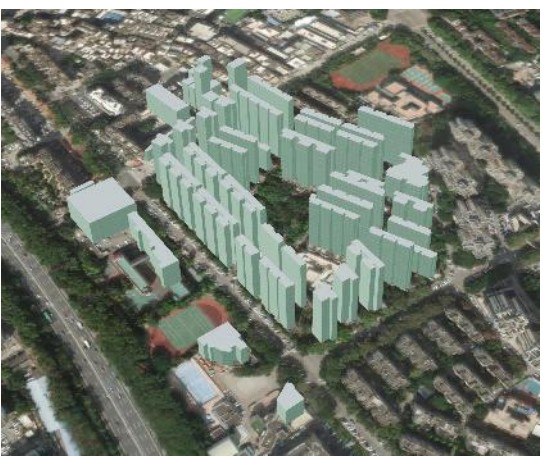
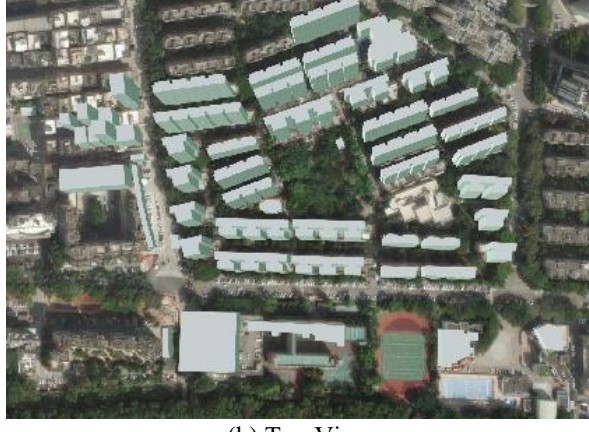

(a) North-East View        (b) Top View

**Figure 4 the building deployment in the case study of the low-rise building cluster, displayed on top of the Bing Map, Copyright of © Microsoft**

It is apparent from Fig. 4 that the case study is a cluster of residential buildings in which both the pedestrian level and mid-

level wind environment could be important with regards of the public health and wind loads acting on the buildings respectively.

---

[3] https://www.ecmwf.int/en/forecasts/datasets/reanalysis-datasets/era5

[4] https://map.baidu.com/

[5] https://gmsh.info/



The WRF simulation of the case study is conducted in two rounds, which includes a preliminary run that provides the initial and boundary conditions for the fine-resolution WRF run, with the Large Eddy Simulation (LES) functionality enabled, of the second round. The configurations of the WRF simulation run for both the preliminary and the second rounds are shown

in Table 1. The simulation results from the second round are actually the trans-scale simulation integrating the WRF-LES run with the OpenFOAM simulation.

**Table 1 the configurations of the WRF model**

| Configurations | WRF (first round) | WRF-LES (second round) |
|---|---|---|
| Time step | 6s | 0.22s |
| Surface layer option | Monin-Obukhov (Janjic) scheme | Monin-Obukhov (Janjic) scheme |
| Land surface option | Unified Noah land-surface model | Unified Noah land-surface model |
| Boundary layer option | Mellor-Yamada-Janjic TKE scheme | Mellor-Yamada-Janjic TKE scheme |
| Urban canopy model | None | Building Environment Parameterization (BEP) scheme |
| WUDAPT LCZ category | Inactive | Activated |
| Eddy coefficient option | horizontal Smagorinsky first order closure | 1.5 order TKE closure (3D) |

The mesh used for the micro-scale simulation are shown in Fig. 5 in two perspectives. In the case study, the PreCICE interface is a box without the lid, which means that the simulation results from the meso-scale model at the top of the micro-

scale domain are not exchanged during the joint simulation. In the trans-scale simulation, the meteorological variables of wind velocities, pressures and turbulent kinetic energies/dissipation rates are exchanged through the coupling interface.

(a) Far-field

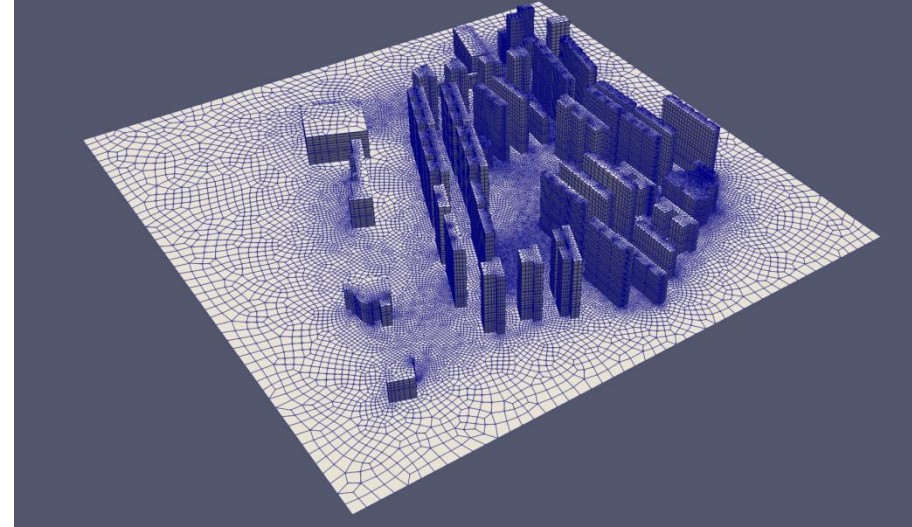





(b) Close-up

**Figure 5 The far-field and close-up views of the OpenFOAM grids of the second case**

The trans-scale simulation is conducted for the time window of 8:00~9:00 on Jan. 20[th] in the year of 2021. While the time step for the meso-scale is approximately 0.22s, the time step of the micro-scale simulation is controlled by the maximum Courant number of 2.0. The configurations of the micro-scale simulation run are shown in Table 2.


**Table 2 the configurations of the OpenFOAM model**

| Configurations | |
| --- | --- |
| Differential scheme for transition term | Crank Nicolson scheme |
| Differential scheme for divergence term | linearUpwind scheme |
| Interpolation scheme | linear |
| Solving algorithm | PIMPLE, the merge of PISO and SIMPLE algorithm |
| Turbulence model | Standard k-epsilon |

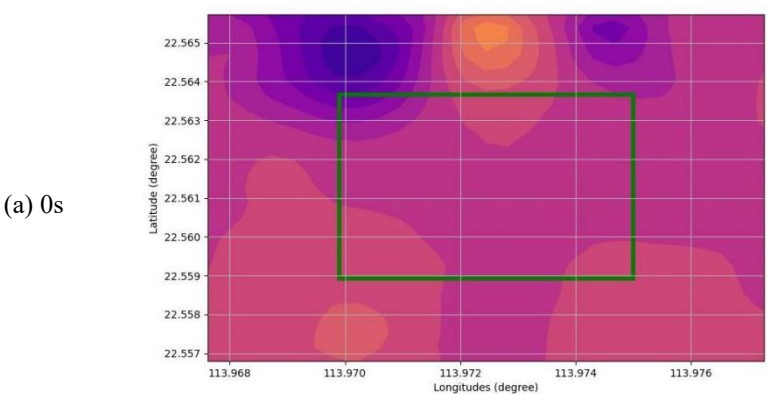

(a) 0s

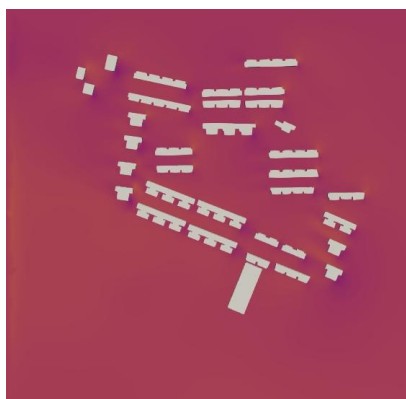





**Figure 6 the comparisons of the south-north wind velocities at the height of 51m from meso-scale and micro-scale models**





**Figure 7 The comparison of west-east wind velocities at the height of 51m from meso-scale and micro-scale models**



Figs. 6 and 7 show the contours of the horizontal components of the wind velocity at the height of 51m from both the meso-scale and micro-scale simulations. For the south-north wind velocity, Fig. 6 shows that the stall zone is well captured by the

micro-scale simulation. Although small fluctuations in south-north wind velocities are observed along its north boundary from the meso-scale simulation results, the trans-scale simulation reveals that the meso-scale wind flow without constant direction poses little influence on the wind environment at the micro-scale. In addition, the densely-placed buildings block the passage of wind flows and create a stall zone within the building group. As indicated in Fig. 7, the passage of the east-west wind flow is largely blocked by the building group and the both horizontal components of the wind velocity within the

circle of the building deployment reduce to approximately zero at the height of 51m.

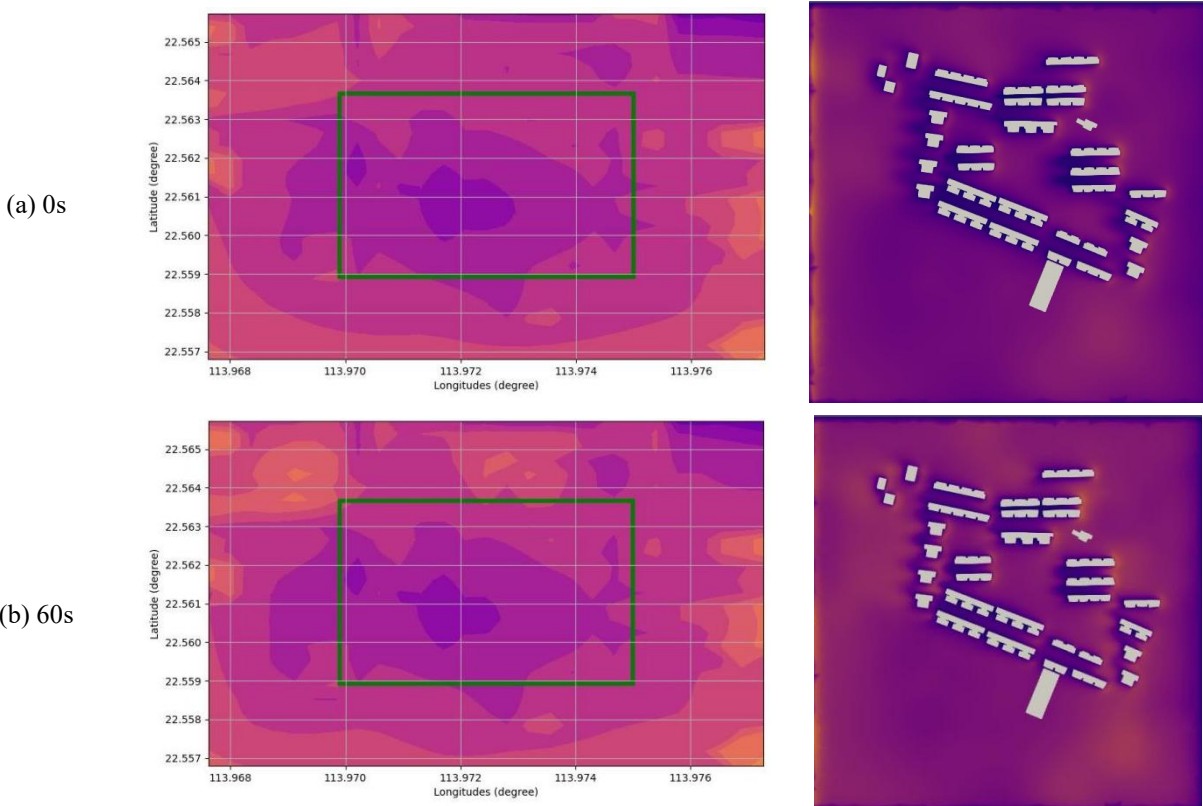



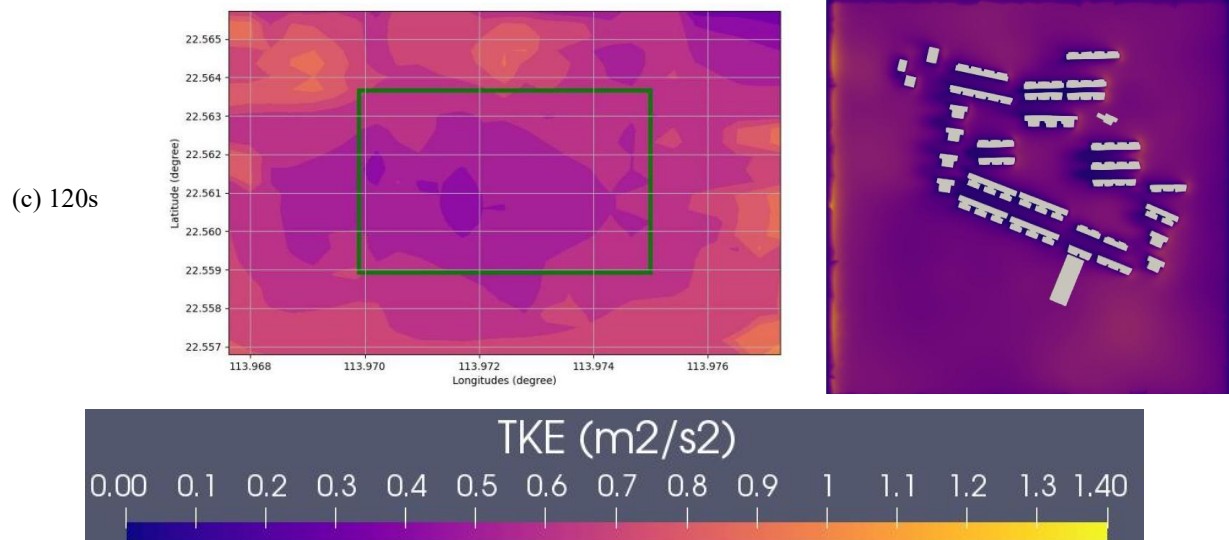

(c) 120s

**Figure 8 The comparison of turbulent kinetic energies at the height of 51m from meso-scale and micro-scale models**

In order to further investigate and compare the wind environments at the mid-height of the building, the turbulent kinetic energies from the meso-scale and micro-scale simulations are plotted as contours in Fig. 8. As in the figures showing the wind velocity comparisons, the comparison of turbulent kinetic energy indicates that the stall zone within the building group corresponds to a low level of turbulence. Since the generation of the turbulent kinetic energies is largely associated with the

interaction between wind flows and building obstacles, such a low level of turbulence is expected. The trans-scale simulation confirms such expectation, and reveals that the meso-scale simulation of the turbulence is, to a certain degree, reliable for predicting the turbulence within the urban canopy layer. Specifically, the comparisons shown in Fig. 8 indicate the urban canopy parameterization activated in the WRF-LES scheme successfully predicts the reduction of the turbulent kinetic energy within the building cluster. The turbulent kinetic energy results along the boundaries of the micro-scale simulation

shown in Fig. 8 are, however, indicates the shortage of the trans-scale simulation. In detail, there are fluctuations in turbulent kinetic energy along the east boundary of the micro-simulation domain, which is not presented by the meso-scale simulation. As the micro-scale simulation requires space for the results to be compatible with its configurations in the one-way nesting scheme, the simulation along the boundaries could be unrealistic in terms of reflecting the interactions between the meso-scale simulation and micro-scale simulation. In other words, the computational domain of the micro-scale simulation should

be artificially expanded to provide the necessary space for the micro-scale simulation to adjust the data provided by the meso-scale model. Such a shortage indicate the direction for the development of the trans-scale simulation framework, which is the inclusion of two-way nesting. As the simulation results feedback into the meso-scale model, the space at the boundaries of the micro-scale simulation domain can be reduced to eliminate the unrealistic simulation results at the boundaries.



## 5 Conclusion

In the investigation of urban wind environment, the numerical simulation tool is vital as it could provide detailed information on both the wind flows and other key meteorological variables, which otherwise requires high costs to acquire through observation campaigns. In the field of geoscience, the numerical simulation also plays an important role as it shows the circulations of large scale atmospheric flows and their realistic interactions with underlying terrains. Along with the growth of the computational capacity and the development of the numerical weather prediction model, it is a trend to integrate the geoscience model into a fine-resolution numerical simulation of the urban wind environment and air quality. When labeled as the meso-scale model (geoscience component) and micro-scale model (urban wind environment simulation tool), there is a relatively long history for researchers to integrate both models to enhance the prediction of the urban wind environment under the influence of specific weather condition.

The present study proposed a framework named WOCSS (version 1.0) to integrate the meso-scale model of WRF and the micro-scale model of OpenFOAM through the newly-emerged tool of PreCICE library. Such integration is capable of simulating the detailed urban wind environment at a block scale of 1km. Thanks to the open-source nature of the WRF and OpenFOAM models, the WOCSS contains a so-called "adapter" to the WRF model and updated the official OpenFOAM "adapter" to handle the communication with the PreCICE library, which takes care of data mapping during coupling and the management of the parallel run of the meso-scale and micro-scale models. Using the reanalysis data available to the public, such framework is capable of not only simulating the detailed wind environment at any given moment in history, but also predicting the urban wind condition in the near future.

Equipped with the WOCSS, the present study shows the trans-scale simulation of densely-placed buildings in a residential quarter in Shenzhen as a case study. The simulation results clearly indicate that the WOCSS integrating the WRF model and the OpenFOAM is successful in terms of reproducing the detailed wind environments in different urban morphology. In fact, the micro-scale simulation of the case study confirms the urban canopy parameterization of the WRF-LES simulation could reliably, to a certain degree, show the reduction in both horizontal wind speeds and turbulent kinetic energies within the stall zone bounded by a group of low-to-median rise buildings. The turbulent kinetic energies close to the boundaries of the micro-scale simulation, however, show nonphysical fluctuations, compared to the meso-scale simulation of the turbulence, which implies the shortage of the one-way nesting adopted in the current version of WOCSS, and hence gives the direction for its version upgrade, i.e. including the two-way nesting functionality in the framework based on the data assimilation already equipped with the WRF model.

## Code Availability

The codes of WOCSS-v1.0 developed by the authors and a tutorial case showing the case study using the WOCSS are organized in the well-established git repository in the site of Gitee (https://gitee.com/skywall/meso-micro-simulation.git) and Github (https://github.com/SunweiTsinghua/meso-micro-simulation.git). In addition, the repository is published via the DOI



of 10.5281/zenodo.7793915. While the installation note is found in the README file of the source code at https://github.com/SunweiTsinghua/meso-micro-simulation/blob/master/README.en.md, the instruction for running WOCSS can be found in the README file of the tutorial case at https://github.com/SunweiTsinghua/meso-micro-simulation/blob/master/tutorial/README

**Author Contribution**

Wei Li :  Software, Writing-original draft preparation;

Shuo Leng: Visualization, Data curation;

Sunwei Li: Conceptualization; Writing-review & editing;

Zhenzhong Hu: Funding acquisition, Project administration;

PakWai Chan: Data curation;

**Acknowledgement**

The work described in this paper was supported by the grants from the Shenzhen Science and Technology innovation Commission (Project No. WDZC20200819174646001) and Guangdong Basic and Applied Basic Research Foundation (Project No: 2022B1515130006). The numerical computations reported in the manuscript is partially performed on Hefei advanced computing centre.

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
