# Peer review of "Introduction of a Trans-scale Numerical Simulation Framework Focusing on Urban Boundary Layer: WOCSS V1.0"

_EGUsphere, 2023_

## Referee Comment (RC1)

**Review of "Introduction of a Trans-scale Numerical Simulation Framework Focusing on Urban Boundary Layer: WOCSS V1.0" by Li et al. (Egusphere-2023-482)**

The manuscript by Li et al. presents the WRF-OpenFOAM Coupled Simulation System (WOCSS) V1.0, which enables one-way nesting from WRF onto a Computational Fluid Dynamics (CFD) model embedded in OpenFOAM incorporated with the PreCICE library. Development of the interfaces between mesoscale and microscale models is a popular and interesting topic with advances in computation resources and the needs of high-resolution numerical simulations. The tool and results presented by the authors are interesting and the manuscript does have its own research value. However, the manuscript lacks detailed explanation regarding the rationale to develop such tool, and the description of WOCSS is also a bit thin. In addition, more analysis on the presented case study is required to demonstrate the performance and values of the tool. The presentation of figures and tables is rather poor and needs improvement. And I would highly recommend English language editing for this manuscript as grammatical errors can be spotted throughout the manuscript, which are quite distracting. Therefore, I would recommend that this manuscript should not be accepted in its current form and a major revision is required. My detailed comments are listed below.

**Major concerns:**

1. The rationale of the WOCSS development was not clear. There are a large number of existing studies that used WRF-OpenFOAM coupling, for example, Temel et al. (2018), Li et al. (2019), and Safaei Pirooz et al. (2021). However, not much literature review regarding such application was provided in the manuscript. I'm aware that OpenFOAM has various models and features while the authors did not state why this study is novel compared to the existing WRF-OpenFAOM studies. What is the difference between WOCSS and the previous studies? Is there any existing tool for WRF-OpenFOAM coupling? These are not explicitly explained in the manuscript.

2. According to the description of the WOCSS framework, WOCSS seems to be an extension of PreCICE that only passes data from WRF to PreCICE and then to OpenFOAM, whereas PreCICE seems to play the major role in the coupling. Can the authors explain the significance of WOCSS development? Why WOCSS is such an important addition to PreCICE? Section 2 may need to be rearranged to highlight the significance of WOCSS.

3. **Section 3.2 Turbulence Estimation:** please provide references for the estimation of the dissipation rate. Can authors clarify why Equation 1 was chosen for the estimation? There are several different methods, for example, refer to Wang et al. (2021) and Beu and Landulfo (2022).

4. **Case study results:** only horizontal cross sections at the elevation of 51 m were presented. More analysis is required to demonstrate the difference between WRF and OpenFOAM and why the usage of WOCSS has any added value. Do the authors have observations in the simulated area? Comparison between the simulations and observations (if any) would be valuable. Also, if there is any existing tool for WRF-OpenFOAM coupling, how is WOCSS different from other tools? The presented results do not provide sufficient information to answer whether WOCSS is valuable or not.

5. **Running meso-scale and micro-scale models in parallel** – What is the advantage? The authors have mentioned several times that most of the current coupling

frameworks do not run meso- and micro-scale models in parallel. The previous studies mostly used an "offline nesting" approach that the meso-scale model finished first and the micro-scale model will take the output from meso-scale. What is the difference between the two approaches (offline vs. in parallel)? In general, a micro-scale model runs slower than a meso-scale model. So without two-way nesting, running in parallel does not seem to provide any advantage. The WOCSS framework presented here is only one-way nesting. There may potentially be values in forecasting or nowcasting at micro-scale? More explanation is required.

**Detailed comments:**

1. Consistency is needed in the wordings. The authors used different wordings when referred to meso-scale or micro-scale **models**, which may cause confusion while reading. For example, the authors referred NWP models as "the NWP **package**" in the beginning of the manuscript; "the Computational Fluid Dynamics (CFD) simulation **code**" in Line 46 should be "Computational Fluid Dynamics (CFD) **models**"; and the authors mentioned "a meso-scale **tool**" in Line 89, while based on the concept of the sentence, it should be "a meso-scale **model**". I would recommend the authors to revise these wordings throughout the manuscript.

2. CFD models are usually computationally expensive. How much computation time did the case study cost? How practical it is to use CFD models for urban climate research?

3. **Line 94:** The authors indicate that the approach used in Bakhoday-Paskyabi et al. (2022) is different from those in Lin et al. (2021) and Kadasch et al. (2021) by stating "Different from such approaches". The threes studies all used the offline nesting approach between a mesoscale model and PALM (models did not run in parallel). Can the authors clarify the difference?

4. **Line 119:** PreCICE was first mentioned here but no references were provided. Recommend adding citation of Chourdakis et al. (2023).

5. **Line 128:** the description of the joint simulation is quite wordy. The details should be presented in Section 2 rather than in Introduction. The authors should only present the key points such that the readers would understand that one-way nesting is used along with WRF, WRF-LES, and the CFD model embedded in OpenFOAM. Please also add references regarding WRF-LES.

6. **Line 145:** the authors mentioned that WOCSS is a "new framework". What is the "old" framework and what is the novelty of this study?

7. **Figure 1:** It is unclear what is the role of WOCSS here. I understand that the flowchart is for running simulations using WOCSS but both the figure and its caption did not mention anything related to WOCSS. This figure can be interpreted as a PreCICE application rather than WOCSS. Based on the description in the main text (Lines 171 to 181), WOCSS passes information from WRF to PreCICE. The resolution of the figure is quite coarse. A better image quality is required.

8. **Line 196:** please specify the variables and don't say "the author simply modified the official adaptor". Details are needed for readers to understand the significance regarding the development of WOCSS.

9. **Figure 2:** consider replotting and the figure caption needs rephrasing. In the figure caption, please mention the subplot labels, namely which panels are for wind velocity and which ones are for TKE. Please add dimension references (x, y, and z) in each panel. Also, the units of plotted variables are missing. Please use figures with better image quality. The figures look like combination of screenshots as the background colours are not uniform, e.g., Figures 2b and 2d look like combination of two screenshots.

10. **Figure 3:** please provide map scales.

11. **Figure 4:** please provide map scales. In the figure caption, please clarify that the blocks in light green are buildings included in the case study. The buildings are not easily distinguishable from the background maps. Recommend using a more recognisable colour.

12. **Table 1:** please add references for the schemes used in WRF simulations.

13. **Line 340:** it would be interesting to outline the grid spacing of WRF simulation domains and the grid spacing or details of the mesh setup in the OpenFOAM simulation. In addition, how did WOCSS interpolate WRF grid onto OpenFOAM grid?

14. **Figure 5:** please provide map scales. Please add figure label descriptions in the figure caption.

15. **Table 2:** please add references for the schemes used.

16. **Figure 6:** Figure caption needs to specify the details of each panel. Please use figures with higher image quality. Please increase the font sizes of axes' labels as they are hard to read in the current form. Please add spatial references of dimensions on all the right-hand side panels. I would recommend using spatial references in metres in all figures rather than using latitudes and longitudes for the purpose of comparison between WRF and OpenFOAM. And in case screenshots were used in this figure, please replot all panels as one uniform figure. What do 0 s, 60 s, and 120 s mean? Clarifications are needed. Same as Figures 7 and 8.

17. **Line 353:** As I stated previously, the comparison between WRF and OpenFOAM was only presented at one vertical level of 51 m for wind velocities. What is the added value of this coupling? The authors need to provide more comparison, such as time series, vertical profiles, and/or hourly composites of winds and temperatures, to show the added value of using OpenFOAM. Do WRF and OpenFOAM results agree with each other? How much more information does OpenFOAM provide?

18. **Figure 8:** There seems to be some artefacts (a band of very high or low values) at the lateral boundaries in OpenFOAM (right hand side subplots). While this is only mentioned in the main text, the authors may want to acknowledge such fluctuations in the figure caption.

19. **Line 366:** "reliable for predicting the turbulence" – how did the authors draw such conclusions? Can the authors provide validation of the model output to prove reliability?

20. **Line 368:** "the urban canopy parameterization activated in the WRF-LES scheme successfully predicts the reduction of the turbulent kinetic energy within the building cluster." It is not surprising that a reduction of TKE is presented when the urban canopy parameterization was enabled. It is however difficult to conclude that the model results are "successful". I would recommend the authors to present comparison between WRF, WRF-LES, OpenFOAM, and observations (if any) to show the improvements and the added values of using simulations with finer scale in OpenFOAM.

21. **Line 376:** "Such a shortage indicates the direction for the development of the trans-scale simulation framework, which is the inclusion of two-way nesting." Any previous studies and references on this statement regarding two-way nesting?

**Minor items:**

1. All figure and table captions – please capitalise the first letter of each sentence.

2. The usage of articles (**the** and **a/an**) across the manuscript needs to be improved. The article **"the"** is used before a noun to indicate that the identity of the noun is known to the reader. For example:
   - Line 14: "CFD simulation tool is the most popular"
   - Line 20: "via the PreCICE library"
   - Line 183: "OpenFOAM is a commonly accepted software"

3. Please revise the usage of a space between the number and an abbreviated unit of measurement in the Abstract and throughout the main text. For example:
   - Line 13: 1 ~ 100 km
   - Line 15: 1 m ~ 1 km
   - Line 103: ~1 m
   - Line 131: ~10 m
   - Line 322: 30 km
   - Line 323: 80 km
   - Line 392: 1 km
4. Line 16: "v1.0" should be "V1.0".
5. Line 17: "thanks to" may not be very formal in scientific writing.
6. Line 25: "geoscience" should be "geoscientific".
7. Line 31: "small-sized turbines" - please specify the size. Remove "also".
8. Line 32-34: Any references?
9. Line 35: "As for" is not a formal way to open a sentence.
10. Line 36: "at the city scale" – please specify the scale (how many meters?).
11. Line 39: "...in physics, numeric, and data assimilation" – should "numeric" be "numerics"?
12. Line 41: Please specify "city scale" and "block scale" so readers don't have to search the references.
13. Line 41: remove "It is noted that".
14. Line 46: "the microscale model, i.e., the Computational Fluid Dynamics (CFD)". As far as I understand, microscale models include CFD, Large Eddy Simulation (LES) model, and other models. This may be a misuse of "i.e.,", which should be "e.g.," instead.
15. Line 51: Remove "actually".
16. Line 66: What does "commonly available" mean? Either a tool is available, or it is not. Or do the authors mean "widely available"?
17. Line 69: Please specify what is "a geophysical scale".
18. Line 70: "utilizes" should be "utilize".
19. Line 71: Please specify "sub-building scale".
20. Line 88: This sentence is wordy. "for coupling the meso-scale model and micro-scale model" and "for running the joint simulation" are essentially the same idea.
21. Line 115: "the mesoscale simulation results at the city scale is..." – here "is" should be "are"
22. Line 120: Please specify "the scale of city blocks".
23. Line 138: "section 2" should be "Section 2".
24. Line 149: Remove "in fact". "the WOCSS simulation" should be "simulations that used WOCSS" because WOCSS is a modelling tool rather than a numerical model.
25. Line 153: Recommend rewording to "The WRF model is well-known for weather forecasting applications…"
26. Line 155-158: please add references for the sentence "The WRF model was developed by… since the 1990s".
27. Line 258: Please specify "the coupling effects". Mentioning of such effect seems abrupt in this section. This usually is included in case studies or discussion sections.
28. Line 275: Why "the horizontal bi-linear interpolation" was chosen for WOCSS?
29. Line 322: "resolution" should be "grid spacing".
30. Lines 322-323: "...ERA5 dataset… in history." Please add references such as Hersbach (2019).
31. Lines 325-335: Recommend having a separate simulation setup section for the descriptions of maps and building outlines.
32. Caption of Figure 4: please add links/references for Bing Maps.

33. Line 347: Is "8:00~9:00" local time? Why was this time chosen for the case study?
34. Line 361: What is "the mid-height of the building"? Does this mean any specific building in the simulation?
35. Line 392: Please revise the use of "Thanks to".

**References:**

Beu, C.M. and Landulfo, E., 2022. Turbulence Kinetic Energy Dissipation Rate Estimate for a Low-Level Jet with Doppler Lidar Data: A Case Study. *Earth Interactions*, *26*(1), pp.112-121.

Chourdakis, G., Schneider, D., & Uekermann, B. (2023). OpenFOAM-preCICE: Coupling OpenFOAM with External Solvers for Multi-Physics Simulations. *OpenFOAM® Journal*, *3*, 1–25. https://doi.org/10.51560/ofj.v3.88

Hersbach, H., 2019. Global reanalysis: goodbye ERA-Interim, hello ERA5. *ECMWF newsletter*, *159*, p.17.

Li, S., Sun, X., Zhang, R. and Zhang, C., 2019. A Feasibility Study of Simulating the Micro-Scale Wind Field for Wind Energy Applications by NWP/CFD Model with Improved Coupling Method and Data Assimilation. *Energies*, *12*(13), p.2549.

Safaei Pirooz, A.A., Moore, S., Turner, R. and Flay, R.G., 2021. Coupling high-resolution numerical weather prediction and computational fluid dynamics: Auckland Harbour case study. *Applied Sciences*, *11*(9), p.3982.

Temel, O., Bricteux, L. and van Beeck, J., 2018. Coupled WRF-OpenFOAM study of wind flow over complex terrain. *Journal of Wind Engineering and Industrial Aerodynamics*, *174*, pp.152-169.

Wang, G., Yang, F., Wu, K., Ma, Y., Peng, C., Liu, T. and Wang, L.P., 2021. Estimation of the dissipation rate of turbulent kinetic energy: A review. *Chemical Engineering Science*, *229*, p.116133.

---

## Author Comment (AC1)

**Feedback to the anonymous reviewer #1**

The authors would like to thank this anonymous reviewer on his/her positive suggestions on the improvements over the original paper. For the the comments provided by the reviewer, the authors give the point-to-point responses as below,

**Major concerns:**

1. The rationale of the WOCSS development was not clear. There are a large number of existing studies that used WRF-OpenFOAM coupling, for example, Temel et al. (2018), Li et al. (2019), and Safaei Pirooz et al. (2021). However, not much literature review regarding such application was provided in the manuscript. I'm aware that OpenFOAM has various models and features while the authors did not state why this study is novel compared to the existing WRF-OpenFAOM studies. What is the difference between WOCSS and the previous studies? Is there any existing tool for WRF-OpenFOAM coupling? These are not explicitly explained in the manuscript.

It is true that a considerable number of previous studies have been conducted to couple the simulations of WRF and OpenFOAM. The most significant advancement of the present study over the methods suggested in the literature is that the spatial and temporal variations in wind velocities and other meteorology variables are explicitly taken into consideration in coupling WRF and OpenFOAM simulations. With regards the studies explicitly mentioned by the reviewer, Temel, et al. (2018) showed the wind flows at Askervein Hill and Ria de Ferrol in Europe via coupling WRF and OpenFOAM. The boundary conditions to run the OpenFOAM simulation were, however, taken from the WRF models via a temporal average process. Pirooz, et al. (2021) coupled the meso-scale model of NZLAM to the CFD simulation with the inlet wind profile of the CFD simulation fitted from the meso-scale model results. Such a feature was also shown in the study of Kadaverugu, et al. (2021) who combined the temporally averaged WRF model results with the OpenFOAM simulation focusing on the urban wind environment of during January 11–18, 2018 over Nagpur City, India. Other than the temporal average, Li, et al. (2019) suggested coupling the WRF model with the OpenFOAM simulation via spatially averaging the WRF simulation results to provide a single wind profile at the inlet boundary of the CFD simulation domain.

The introduction of spatial and temporal variations in the data exchange between the meso-scale and micro-scale model make use of the full capacity of the micro-scale model in terms of enriching the space-time structure of the large eddies from the meso-scale simulation at the cost of dealing with data mapping and temporal interpolation at every time step for the micro-scale simulation. This is where the WOCSS stands up to play a significant role. For one thing, the spatial and temporal variations retained in the data passed to the micro-scale model at its boundary let the micro-scale model to resolve the detailed structures inside the urban boundary layer with higher resolution mesh and more advanced turbulence close schemes. For another thing, the retention of spatial and temporal variation in the data exchange requires the wind velocity simulated by the meso-scale model with relatively coarse mesh mapped onto the micro-scale model with high resolution mesh every time step. In such a case, the methods used in the previous studies (e.g. Pirooz, et al. (2021) and    Kadaverugu, et al. (2021)) involving a lot of manual adjustment and supervised spatial interpolation are no longer applicable. The automatic and robust algorithms, and corresponding implementation, shown by the WOCCS therefore are helpful and valuable for

conducting the coupled simulation to show the detailed space-time structures in the urban boundary layer wind field. In addition, the retention of the temporal variation means that the wind fields from the meso-scale simulation should be interpolated to drive the micro-scale simulation at the micro-scale time step. This is also the place where the WOCSS to shine as previous algorithms are usually only capable of coping with the quasi-steady state at the micro-scale side. In these so-called "snapshot" approaches, the temporal variations in the results from the meso-scale simulation are filtered with a very-large time scale. Therefore, the temporal interpolation is often neglected in the "snapshot" approach, and it is assumed that the simulation results from the micro-scale side at the previous moment (previous snapshot) has no impacts on the current micro-scale simulation. To illustrate the differences, the micro-scale simulation of the case study is repeated using the results from the meso-scale simulation at the starting moment of the coupled simulation. In other words, the "snapshot" simulation, following the methodology of Temel, et al. (2018), is conducted to compare the coupled simulation results. In addition, the results from the meso-scale simulation are fitted to the well-known profiles as suggested by Li, et al. (2019) to drive the simulation at the micro-scale side. The three simulations will be termed as "coupled", "snapshot", "profile-fitted" simulations, and their results corresponding to 60s after the starting moment of the coupled simulation are compared in Figure 1.

The manuscript will be revised accordingly to emphasize such an contribution of the present study.

[Figure]

| | |
|---|---|
| profile-fitted |
[Figure]
 |
| Figure 1 the comparison of the east-west wind velocity at the height of 2m | |

2. According to the description of the WOCSS framework, WOCSS seems to be an extension of PreCICE that only passes data from WRF to PreCICE and then to OpenFOAM, whereas PreCICE seems to play the major role in the coupling. Can the authors explain the significance of WOCSS development? Why WOCSS is such an important addition to PreCICE? Section 2 may need to be rearranged to highlight the significance of WOCSS.

It is true that the existing PreCICE library take the major responsibility for coupling the meso-scale and micro-scale simulation, mainly in the fields of data mapping and simultaneous communications between both sides. However, the PreCICE library is a set of general-purpose codes to run the coupled simulation. It is the "adapter" to various existing numerical simulation tools that makes the PreCICE library practically useful. Considering the wide application of the meteorology simulation code of WRF, the WOCSS essentially presents the "adapter" to WRF in the framework of PreCICE. For one thing, WOCSS make the WRF collaborate with other numerical tools in the simulation of cross-scale flow fields, heat transfers and other meteorological process. For example, the WRF simulation results could be used to derive ENVI-met for the assessment of the urban heat island effect via a similar approach as the WOCSS. For another thing, the WOCCS also widen the application of the PreCICE library. Currently, PreCICE mainly support the common simulation tools of fluid dynamics, structural dynamics and thermodynamics. As the WOCSS introduces the coupling of a meteorology package, other numerical simulations already built upon the PreCICE library sees the opportunity to run coupled simulation with the meteorology package.

The manuscript will be revised accordingly to illustrate the significance of the implementation of WOCSS, with regards to both the WRF model and the PreCICE library.

3. Section 3.2 Turbulence Estimation: please provide references for the estimation of the dissipation rate. Can authors clarify why Equation 1 was chosen for the estimation? There are several different methods, for example, refer to Wang et al. (2021) and Beu and Landulfo (2022).

The turbulence dissipation rate could surely be estimated from the meso-scale simulation results according to various models, and it is true that the calculation presented in the original manuscript is not superior when comparing to other available methods. Consequently, the WOCCS itself is recoded to provide the end-user the options to estimate the turbulence dissipation rate. In fact, the namelist file controls the behaviour of the WOCSS side is amended to take the option in estimating various turbulent variables. Currently, the WOCSS supports, (a) the conventional approach which

utilizes the turbulent kinetic energy and length scales as in equation (1); (b) the method suggested by   which requires the WRF simulation to run with certain planetary boundary layer schemes and (c) the method in accordance with several selections in the review of Wang, et al (2021), which requires the WRF simulation provides the results of turbulent kinetic energy and different types of turbulence length scales.

4. Case study results: only horizontal cross sections at the elevation of 51 m were presented. More analysis is required to demonstrate the difference between WRF and OpenFOAM and why the usage of WOCSS has any added value. Do the authors have observations in the simulated area? Comparison between the simulations and observations (if any) would be valuable. Also, if there is any existing tool for WRF-OpenFOAM coupling, how is WOCSS different from other tools? The presented results do not provide sufficient information to answer whether WOCSS is valuable or not.

The authors acknowledged that the value of the proposed WOCSS is not fully illustrated in the present study. More specifically, the intention of the case study reported in the original manuscript is merely showing the feasibility of running the WRF and OpenFOAM coupled simulation for a target urban area, not to show the advantages of the WOCSS. Therefore, more simulation results will be reported in the revised manuscript.

It should be noted that the accuracy of the coupled simulation inside a residential quarter relies on a variety of factors, and is not the focus of the present study. For example, the simulation at the meso-scale side relies on the planetary boundary layer and urban canopy modelling to show the realistic wind flow in the urban area. The simulation at the micro-scale side, on the other hand, are strongly influenced by the numeric differentiation schemes and turbulence models. Consequently, the comparison to the observations reveals the overall effect of the combination of various factors, and the advantages of the WOCCS could be buried. In fact, the value of the WOCSS is to provide the information otherwise not available from both the meo-scale side and the mciro-scale side. For one thing, the meso-scale simulation does not yields the detailed wind field explicitly considering the geometries of the buildings, and therefore the dangerous wind speeds threatening the safety of the building attachments, such as the speed-up effect around the building corner, are not available. For another thing, the micro-scale simulation, which yields the information with sufficient details for the assessment of building performance and pedestrian comforts, frequently run without realistic boundary conditions, which leads to unrealistic estimates of micro-scale wind field.

Therefore, the revised manuscript contains the following comparison for the purpose of substantiates the value of the WOCSS. (a) the wind fields resulted solely from the the meso-scale simulation and from the WOCSS around a building corner are compared to show that the WOCSS is able to present the corner flow due to the blocking effect of the building; (b) the wind field resulted from the OpenFOAM simulation in a steady-state with the inlet boundary conditions specified as a logarithmic profile model whose parameters are fitted from the meso-scale simulation is compared to the simulation of WOCCS, and the comparison indicates that the profile fitting, such as in the study of Li, et al. (2019), is insufficient to specify the inlet boundary conditions of the OpenFOAM if the detailed wind flow at the mid-height of the building is of interest; (c) the wind field generated by the steady-state OpenFOAM simulation with boundary conditions specified by the temporal averaged wind flow from the meso-scale mdoel is compared to the WOCSS simulation results, which reveals the effect to include the temporal variation in the WRF-OpenFOAM coupled

simulation. The simulated wind velocity magnitudes at the 75s from the starting moment of the coupled simulation are shown in Figure 2.

[Figure]

Figure 2 the comparison of the wind velocity magnitude at the height of 2m

5. Running meso-scale and micro-scale models in parallel – What is the advantage? The authors have mentioned several times that most of the current coupling frameworks do not run meso- and micro-scale models in parallel. The previous studies mostly used an "offline nesting" approach that the meso-scale model finished first and the micro-scale model will take the output from meso-scale. What is the difference between the two approaches (offline vs. in parallel)? In general, a micro-scale model runs slower than a meso-scale model. So without two-way nesting, running in parallel does not seem to provide any advantage. The WOCSS framework presented here is only one-way nesting. There may potentially be values in forecasting or nowcasting at micro-scale? More explanation is required.

To be completely honest, the parallel coupling with one-way nesting configuration only provides limited advantages comparing to the offline nesting. For the offline nesting with a relatively high resolution of the coupling interface, the data generated from the meso-scale simulation could be large if the exchange occurs after the meso-scale simulation run is finished. In such a case, the offline nesting takes additional space in hard drive comparing to the parallel coupling. In addition, reading the large data at the coupling interface into the micro-scale model takes additional computational time via communicating with the hard drive during the simulation or requires large memory to store all data at the beginning.

The true value of the parallel coupling, however, is the possibility it provided to run the two-way nesting coupled simulation. With the parallel coupling implemented in the present study, the management of running two processes corresponding to the meso-scale and micro-scale model simultaneously, and the communication between them are solved. More specifically, the mechanism for blocking one process and waiting for the coupled data and communications between two processes using the I/O widely supported by the Linux system have all been implemented. Therefore, the two-way nesting coupling, which will be the development target of the WOCSS 2.0, will be ready once the revisions are made to the source code of the WRF model to properly spread the data from the OpenFOAM simulation into its own computational domain.

**Detailed comments:**

1. Consistency is needed in the wordings. The authors used different wordings when referred to meso-scale or micro-scale models, which may cause confusion while reading. For example, the authors referred NWP models as "the NWP package" in the beginning of the manuscript; "the Computational Fluid Dynamics (CFD) simulation code" in Line 46 should be "Computational Fluid Dynamics (CFD) models"; and the authors mentioned "a meso-scale tool" in Line 89, while based on the concept of the sentence, it should be "a meso-scale model". I would recommend the authors to revise these wordings throughout the manuscript.

Thank you for your constructive suggestion, the wording are revised accordingly. The revised manuscript uses the "meso-scale model" and "micro-scale model" throughout the paper except for the very beginning to point out that the "meso-scale model" is in most cases the NWP package in the field of meteorology while the "micro-scale model" is commonly the CFD simulation code.

2. CFD models are usually computationally expensive. How much computation time did the case study cost? How practical it is to use CFD models for urban climate research?

The reviewer is right about the computational cost of the CFD simulation reported in the present

study is still high for practical use. In fact, the computational time of the case study using the personal computer of the authors is in the order of 10 hours. This computational cost make it inappropriate to predict the air ventilation under the emergency (such as in the case of hazardous gas leak). However, the computational cost is acceptable for research on the influence of detailed urban morphology on the localized wind field under the influence of various weather conditions. For example, the WOCCS can be used to retrace the causes of cladding damages after a strong wind event.

3. Line 94: The authors indicate that the approach used in Bakhoday-Paskyabi et al. (2022) is different from those in Lin et al. (2021) and Kadasch et al. (2021) by stating "Different from such approaches". The threes studies all used the offline nesting approach between a mesoscale model and PALM (models did not run in parallel). Can the authors clarify the difference?

The studies of Lin et al. (2021) and Kadasch et al. (2021) only suggested the ways to read the wind data from the meso-scale simulation, and superimpose the artificial turbulence via eddy synthesizing, which could be used as inlet boundary conditions for the micro-scale simulation, but not involves the coupled simulation. In contrast, the study of Bakhoday-Paskyabi et al. (2022) presented the coupled simulation with the meso-scale model of WRF and the micro-scale model of PALM. In addition, the studies of Lin et al. (2021) and Kadasch et al. (2021) targets the micro-scale model of COSMO, but PALM. The text here is misleading, and will be revised.

4. Line 119: PreCICE was first mentioned here but no references were provided. Recommend adding citation of Chourdakis et al. (2023).

The authors acknowledge this is a flaw, and the suggested reference is added.

5. Line 128: the description of the joint simulation is quite wordy. The details should be presented in Section 2 rather than in Introduction. The authors should only present the key points such that the readers would understand that one-way nesting is used along with WRF, WRF-LES, and the CFD model embedded in OpenFOAM. Please also add references regarding WRF-LES.

The text will be abbreviated in the revised manuscript with proofreading done by a native speaker. The reference concerning the WRF-LES will be added.

6. Line 145: the authors mentioned that WOCSS is a "new framework". What is the "old" framework and what is the novelty of this study?

The major novelty of the WOCSS framework is to introduce (a) spatial and temporal variations in wind velocities in the process of data exchanging between meso-scale and micro-scale models; and (b) running of WRF and OpenFOAM processes are in parallel with widely adopted platform of PreCICE, which enables the two-way nesting in the future. Along with the growth in the use of PreCICE, the WOCSS could go with the PreCICE

7. Figure 1: It is unclear what is the role of WOCSS here. I understand that the flowchart is for running simulations using WOCSS but both the figure and its caption did not mention anything related to WOCSS. This figure can be interpreted as a PreCICE application rather than WOCSS. Based on the description in the main text (Lines 171 to 181), WOCSS passes information from WRF to PreCICE. The resolution of the figure is quite coarse. A better image quality is required.

The figure is not clear for presenting the general architecture of the WOCSS, and will be redraw in the revised manuscript.

8. Line 196: please specify the variables and don't say "the author simply modified the official adaptor". Details are needed for readers to understand the significance regarding the development of WOCSS.

The text will be reworded in the revised manuscript and the details for the "adapter" to take into more variables (such as turbulent kinetic energy and dissipation rate) and processed for driving the OpenFOAM simulation is presented.

9. Figure 2: consider replotting and the figure caption needs rephrasing. In the figure caption, please mention the subplot labels, namely which panels are for wind velocity and which ones are for TKE. Please add dimension references (x, y, and z) in each panel. Also, the units of plotted variables are missing. Please use figures with better image quality. The figures look like combination of screenshots as the background colours are not uniform, e.g., Figures 2b and 2d look like combination of two screenshots.

The figure caption will be revised to include the descriptions of the variables of the subplots. The dimensions will be easily added to the figure as they are plotted with the software of Paraview. In addition, the units of the variables will be added to the title of the color bar of the figure as shown in this response. The quality of the figure could be damaged by the journal online submission system converting the MS WORD file into the PDF file. The figures with higher resolution will be tried to upload to the submission system.

10. Figure 3: please provide map scales.

The map scale will be added to the figure.

11. Figure 4: please provide map scales. In the figure caption, please clarify that the blocks in light green are buildings included in the case study. The buildings are not easily distinguishable from the background maps. Recommend using a more recognisable colour.

The map scale will be provided in the revised manuscript, and it will be clearly stated that the blocks shown in the figures are the buildings used for the micro-scale simulation. The buildings will be colored white in the revised manuscript.

12. Table 1: please add references for the schemes used in WRF simulations.

It is a mistake the authors have made, the appropriate references will be added in the revised manuscript.

13. Line 340: it would be interesting to outline the grid spacing of WRF simulation domains and the grid spacing or details of the mesh setup in the OpenFOAM simulation. In addition, how did WOCSS interpolate WRF grid onto OpenFOAM grid?

The grid spacing of the meso-scale model will also be summarized in Table 1 in the revised manuscript. Because the mesh is not equal-distance at the micro-scale side, the statistics of the mesh, including the grid spacing extremes will be contained in the revised manuscript. The data mapping, which interpolates the data from the WRF simulation to the OpenFOAM simulation, is controlled

by the PreCICE library, and a number of different interpolation schemes are available. The corresponding configuration will be added to the revised manuscript.

14. Figure 5: please provide map scales. Please add figure label descriptions in the figure caption.

The dimensions of the buildings will be presented in the revised manuscript and the description of the subplots are added to the figure caption.

15. Table 2: please add references for the schemes used.

It is a mistake of the authors, the appropriate references will be added to the revised manuscript.

16. Figure 6: Figure caption needs to specify the details of each panel. Please use figures with higher image quality. Please increase the font sizes of axes' labels as they are hard to read in the current form. Please add spatial references of dimensions on all the right-hand side panels. I would recommend using spatial references in metres in all figures rather than using latitudes and longitudes for the purpose of comparison between WRF and OpenFOAM. And in case screenshots were used in this figure, please replot all panels as one uniform figure. What do 0 s, 60 s, and 120 s mean? Clarifications are needed. Same as Figures 7 and 8.

While the figures will be replotted as in the previous figures, the dimensions of the computational domain are added to the right panel of the figure. Using the same coordinates, the simulation results from the WRF simulation could be redrawn to show the spatial association between the WRF and the OpenFOAM results. For the time label, the actual time, not the time elapsed from the start of the coupled simulation are used.

17. Line 353: As I stated previously, the comparison between WRF and OpenFOAM was only presented at one vertical level of 51 m for wind velocities. What is the added value of this coupling? The authors need to provide more comparison, such as time series, vertical profiles, and/or hourly composites of winds and temperatures, to show the added value of using OpenFOAM. Do WRF and OpenFOAM results agree with each other? How much more information does OpenFOAM provide?

As responded above, the value of the coupled simulation will be illustrated in the revised manuscript as comparisons (1) the comparison between sole WRF simulation and the coupled simulation, focusing on the corner flow of the building; (2) the WOCSS simulation with the micro-scale simulation using wind profile fitted from the meso-scale results as the inlet boundary condition; (3) the WOCSS simulation with the coupled simulation ignoring the temporal variation.

18. Figure 8: There seems to be some artefacts (a band of very high or low values) at the lateral boundaries in OpenFOAM (right hand side subplots). While this is only mentioned in the main text, the authors may want to acknowledge such fluctuations in the figure caption.

The advice of the review will be accepted in the revised manuscript.

19. Line 366: "reliable for predicting the turbulence" – how did the authors draw such conclusions? Can the authors provide validation of the model output to prove reliability?

Such a conclusion is drawn based on that the micro-scale simulation uses more sophisticated turbulence model and finer mesh to capture the turbulent variations in the urban canopy layer. Therefore, it is reasonable to acknowledge that the turbulent kinetic energy from the micro-scale

simulation is more reliable. Consequently, the agreement between the meso-scale model and micro-scale model reveals the reliability of the meso-scale model in simulating urban canopy turbulence.

20. Line 368: "the urban canopy parameterization activated in the WRF-LES scheme successfully predicts the reduction of the turbulent kinetic energy within the building cluster." It is not surprising that a reduction of TKE is presented when the urban canopy parameterization was enabled. It is however difficult to conclude that the model results are "successful". I would recommend the authors to present comparison between WRF, WRF-LES, OpenFOAM, and observations (if any) to show the improvements and the added values of using simulations with finer scale in OpenFOAM.

Considering the absence of the observation in the residential quarter under investigation in the present study, the simulation of the meso-scale model is compared to the simulation of the micro-scale model concerning the corner flow of the buildings. Such a feature is widely acknowledged in the building aerodynamics and in the assessment pedestrian level comfort. Since the WRF and the WRF-LES utilizes parameterization to artificially increases turbulence level according to the density and averaged height of the building cluster, the successful capture of the high level turbulence not only indicates the activation of urban canopy parameterization in WRF is necessary to evaluate the turbulence in the urban boundary layer, but also reveals the default database used by the WRF to describe the urban morphology worldwide is accountable.

21. Line 376: "Such a shortage indicates the direction for the development of the trans scale simulation framework, which is the inclusion of two-way nesting." Any previous studies and references on this statement regarding two-way nesting?

Such a shortage is spotted out in the present study. More specifically, previous studies emphasize the importance of including the two-way nesting in the coupled simulation from the perspective of providing more accurate simulation results at the meso-scale side. The present study, however, indicates that the two-way nesting also benefits the micro-scale simulation in terms of improving the quality of the boundary conditions for the micro-scale simulation.

**Minor items:**

The authors would like to thank this anonymous reviewer for the correction suggestions in grammar and language, those suggestions will be taken in the revised manuscript. Following are the revisions that does not directly pertain to the language.

1. Line 31: "small-sized turbines" - please specify the size. Remove "also".

The turbine is installed on top of urban buildings, and therefore the installed capacity ranges from 1KW ~ 100KW. However, there has been continuous discussion on the installation of multi-MW wind turbines.

2. Line 32-34: Any references?

Appropriate references will be added to the revised manuscript.

3. Line 35: "As for" is not a formal way to open a sentence.

4. Line 36: "at the city scale" – please specify the scale (how many meters?).

There are generally three different scales in the numerical simulations reported in the present study: (a) city scale, usually at the scale of 10km and above; (b) block scale, usually at the scale of 100m ~ 1km; and (c) building scale ~1m. Those information will be added to the revised manuscript.

5. Line 46: "the microscale model, i.e., the Computational Fluid Dynamics (CFD)". As far as I understand, microscale models include CFD, Large Eddy Simulation (LES) model, and other models. This may be a misuse of "i.e.,", which should be "e.g.," instead.

Not exactly. The authors considers the Large Eddy Simulation a branch of the CFD simulation. In other words, the micro-scale model generally corresponds to the CFD simulation, different models includes different additional variables and uses different turbulence models within the framework of the CFD simulation.

6. Line 69: Please specify what is "a geophysical scale".

This is the city scale, it will be corrected and the the information on different scales used in the present simulation are provided in the revised manuscript.

7. Line 71: Please specify "sub-building scale".

As the building scale is in the order of 1m, the sub-building scale is below 1m. The expression will be clarified in the revised manuscript.

8. Line 155-158: please add references for the sentence "The WRF model was developed by… since the 1990s".

The appropriate references will be added to the revised manuscript.

9. Line 275: Why "the horizontal bi-linear interpolation" was chosen for WOCSS?

This is because it is simple to implement, and the more advanced interpolation schemes could be developed in the future.

10. Lines 322-323: "...ERA5 dataset… in history." Please add references such as Hersbach (2019).

Thanks for the suggestion, the appropriate references are added.

11. Lines 325-335: Recommend having a separate simulation setup section for the descriptions of maps and building outlines.

The suggestion is taken and the description of the simulation set-ups and other relevant information is gathered in a separate section.

12. Caption of Figure 4: please add links/references for Bing Maps.

The appropriate link is added to the caption

13. Line 347: Is "8:00~9:00" local time? Why was this time chosen for the case study?

This is the local time, the time is chosen randomly as it is for the demonstration of the capability of the WOCSS framework, not to investigate the urban wind environment under the influence of specific weather conditions.

14. Line 361: What is "the mid-height of the building"? Does this mean any specific building in the

simulation?

This is generally at the mid-height of the building cluster, as the majority of the buildings are at the height of ~100m (30-40 stories)

---

## Author Comment (AC2)

**Feedback to the anonymous reviewer #2**

The authors would like to thank this anonymous reviewer on his/her comments on the manuscript, and provide the point-to-point responses to his/her comments as follows,

There is also a general lack of details regarding model description, test case preparation and discussion on preliminary results. Based on the information presented in the MS, I do not get the sense that the authors have performed significant original work on this coupling framework to warrant a standalone publication.

Although the authors recognizes that the present study is built upon the existing coupling library of PreCICE, the effort is still worth for a standalone publication. This is because

a. It is the adaption made to the source code of various existing numerical models that makes the PreCICE library a useful tool. Therefore, the adaptions made to both widely-used numerical models of WRF and OpenFOAM extends the applicability of PreCICE and WRF/OpenFOAM. In fact, the WOCSS proposed in the present study can be used in simulating the wind flow, predicting air qualities and retracing the source of pollutants with much finer resolution.

b. The existing studies coupled the runs of WRF and OpenFOAM often ignores either the temporal or the spatial variations in the wind field, which makes only a small fraction of the simulation data are exchanged at the coupling interface. The present study resembles the attempt to include the spatial and temporal variations at the scales corresponding to the finest grid of the meso-scale model in coupling the WRF and OpenFOAM, which could be the foundation for the development of a numerical weather prediction tool dealing with the flow at the scales of atmospheric circulation down to the viscous flow near the building surface.

c. The complex architecture of the WRF model, which contains more than 100000 lines of source code, make the adaption to include the communication with PreCICE a challenging task. In fact, the authors have continuously made efforts to extract the required information to drive the micro-scale simulation for a couple of years. The experience with running the WRF simulation and general coding finally enables the authors to spot out the piece of source codes inside the WRF model to be adapted. In addition, debugging, documentation and finalization of the source code based on a publicly available git repository makes it a piece of work worthwhile to be published in a journal like GMD

To illustrate the influences of including the temporal variations in wind field into the couped simulation, the micro-scale simulation of the case study is repeated with the boundary conditions determined by the meso-scale simulation results at the moment starting the coupled simulation (termed as snapshot simulation). The simulated wind fields at the 75s from the starting moment are shown in Figure 1.

[Figure]

Figure 1 the comparison of the wind velocity magnitude at the height of 2m

In addition, it is not intersting to reiterate OpenFOAM solver settings (viz. Table 2) without indicating which OpenFOAM solver is being used (I assume the default solver simpleFoam was used), or to show a mesh of the urban geometry (viz. Fig 5) without providing sufficient context on how the mesh is generated or the mesh statistics (i.e., how many cells, what type of cells, and mean/max/min cell size).

It is indeed the drawbacks of the original manuscript as the solver of the OpenFOAM package and the statistics of the mesh influences the reliability and the accuracy of the simulation results. In fact, the pimpleFoam, not the simpleFoam, is used for the simulation at the micro-scale side and will be explicitly articulated in the revised manuscript. In addition, the statistics of the CFD mesh will be added to the revised manuscript as well.

Description of the parent mesoscale model is equally sparse, as with the configurations for the PreCICE coupling scheme.

Since the WRF model is widely applied in the field of meteorology in simulating the atmospheric flow at the city scale (~10km), the configuration to run the WRF simulation is attenuated into a Table. Considering the popularity of the WRF model, the WRF-LES simulation scheme will be

emphasized in the revised manuscript to include the origin, development and a few exemplar application of the WRF-LES simulation. In addition, the configuration of the PreCICE in the case study will be summarized in another table in the revised manuscript, including the schemes for data mapping and temporal interpolation.

Further, showing that the OpenFOAM simulation can run for 120 seconds (Figs 6-8) at a qualitative level is far adequate. For instance, Chan and Butler (2021) and Piroozmand et al (2020) devised coupling frameworks for OpenFOAM, both capable of operating for at least a 24 hour period.

For the study of Chan and Butler (2021), the simulation is conducted with an idealized street canyon, whose geometry is simple enough to reduces the size of the mesh. More importantly, the boundary conditions, or the free stream velocity in their study, are specified as constant. Consequently, they are the "snapshot" simulations corresponding to a few specified moments, in which the temporal variations in the meteorological wind field is entirely neglected. In other words, the results from their OpenFOAM simulation do not contain the large scale variations but only shows the detailed dynamic patterns of the flow inside a street canyon under the influence of steady meteorology forcing. Such a methodology could be helpful in assessing the flow pattern and pollution dispersion mechanism around an idealized urban morphology, but is not applicable to forecast the detailed wind field within a urban block as in the present study.

For the study of Piroozmand et al (2020), it is much like the simulation reported in the present study expect that the regional meteorology model of COSMO is used instead of the meso-scale model of WRF. The main difference between the two studies are,

a.  The domain sizes of the regional meteorology simulation for COSMO and WRF are different, while the COSMO simulation shows the domain size of 25km, the WRF simulation reported in the present study gives the domain in the length of 100km. In other words, the COSMO simulation requires the initial and boundary conditions from another large scale simulation but the WRF could take the data from a global model to initialize and to bound its simulation.

b.  The mesh corresponding to the CFD simulation of Piroozmand et al (2020) is relatively coarse comparing to the mesh reported in the present study. More specifically, the number of grids for a dense area of Zurich is in the order of 4 million, while the number of grids for a residential quarter is above 5 million in the present study. Therefore, the detailed flow structures around the building corner, which could be absented from the simulation of Piroozmand et al (2020) are reported in the present study.

c.  The coupling scheme is already implemented within the framework of PreCICE in the present study while the coupling between OpenFOAM simulation and the COSMO simulation requires expert experiences and hence involves manual twisting. In other words, the development and publish of WOCSS enable the scholars who is interested in running the coupled simulation could download the source code of WOCCS and do it themselves following the instructions available on the website running the case study reported in the present study.

While reading Section 3.1 I have understood that the Authors used PreCICE to create a 3D field from the WRF model data, which is then prescribed into the OpenFOAM domain. This approach is incorrect,

as this will cause spurious solution fields in the urban canopy region, in addition of not being mass or energy conservative. An acceptable approach is to map the WRF variable fields as lateral boundary conditions for the OpenFOAM domain, which allows the solution field inside the OpenFOAM domain to develop in its own accord. Of course, this is technically much more challenging.

The original text could be misleading for running the coupled simulation. The WOCSS actually maps the results from the WRF simulation at the lateral and top boundaries to pose the large-scale forcing on the micro-scale simulation. In other words, the WOCSS actually coincides with the suggestions from this reviewer and agrees with the study of Jeanjean et al (2015). The creation of various 3D fields based on the meso-scale simulation is only used to initialize the simulation at the micro-scale side to shorten the spin-up time of the coupled simulation. In fact, the end-user has the option to enable this feature to initialize the OepnFOAM simulation or simply have the zero wind velocity as the initial conditions of the OpenFOAM simulation.